# Modular Gaussian Processes for Transfer Learning

**Pablo Moreno-Muñoz**[*]    **Antonio Artés-Rodríguez**[†]    **Mauricio A. Álvarez**[‡]

[*]Section for Cognitive Systems, Technical University of Denmark (DTU)
[†]Dept. of Signal Theory and Communications, Universidad Carlos III de Madrid, Spain
[†]Evidence-Based Behavior (eB2), Spain
[‡]Dept. of Computer Science, University of Sheffield, UK
pabmo@dtu.dk, antonio@tsc.uc3m.es, mauricio.alvarez@sheffield.ac.uk

## Abstract

We present a framework for transfer learning based on modular variational Gaussian processes (GP). We develop a module-based method that having a dictionary of well fitted GPs, one could build ensemble GP models without revisiting any data. Each model is characterised by its hyperparameters, pseudo-inputs and their corresponding posterior densities. Our method avoids undesired data centralisation, reduces rising computational costs and allows the transfer of learned uncertainty metrics after training. We exploit the augmentation of high-dimensional integral operators based on the Kullback-Leibler divergence between stochastic processes to introduce an efficient lower bound under all the sparse variational GPs, with different complexity and even likelihood distribution. The method is also valid for multi-output GPs, learning correlations a posteriori between independent modules. Extensive results illustrate the usability of our framework in large-scale and multi-task experiments, also compared with the exact inference methods in the literature.

## 1 Introduction

Imagine a supervised learning problem, for instance regression, where $N$ data points are processed for training a model. At a later time, new data are observed, this time corresponding to a binary classification task, that we know are generated by the same phenomena, e.g. using a different sensor.

Having kept the observations from regression stored, a common approach would be to use them in combination with the classification dataset to generate a new model. However, this practice might be inconvenient because of *i)* the need of centralising data to train the model, *ii)* the rising data-dependent computational cost as the number of samples increases and *iii)* the obsolescence of fitted models, whose future usability is not guaranteed for the new set of observations. Looking at the deployment in large-scale scenarios, by any organization or use case, where data are ever changing, this solution becomes prohibitive. The main challenge is to incorporate unseen tasks as former models are recurrently discarded.

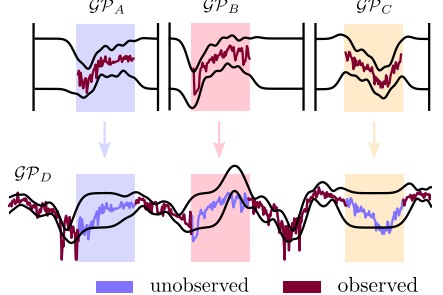

Figure 1: GP modules (*A*, *B*, *C*) are used for training (*D*) without revisiting data.

Alternatively, we propose a framework based on *modules* of Gaussian processes (GP) (Rasmussen and Williams, 2006). Considering the current example, the regression model (or *module*) is kept intact. Once new data arrives, one fits a *meta*-GP using the module, but without revisiting any sample. If no data are observed, combining multiple modules is also allowed — see Figure 1. Under this framework, a new family of module-driven GP models emerges.

35th Conference on Neural Information Processing Systems (NeurIPS 2021).

**Background.** Much recent attention has been paid to the notion of efficiency, e.g. in the way of accessing to data. The mere limitation of data availability forces learning algorithms to derive new capabilities, such as, distributing the data for *federated learning* (Smith et al., 2017), recurrently observe streaming samples for *continual learning* (Goodfellow et al., 2014) and limiting data exchange for *private-owned models* (Peterson et al., 2019). A common theme is the idea of model memorising and recycling, that is, using the already fitted parameters for an additional task without revisiting any data. If we look to the probabilistic view of this idea, uncertainty is harder to be repurposed than parameters of functions. This is the point where Gaussian processes play their role.

The flexible nature of GPs for defining a prior distribution over non-linear function spaces has made them a suitable alternative in probabilistic regression and classification. However, GP models are not immune to settings where we need to adapt to *irregular* ways of observing data, e.g. non-centralised observations, asynchronous samples or missing inputs. Such settings, together with the well-known computational cost of GPs, typically $\mathcal{O}(N^3)$, has motivated plenty of works focused on *parallelising inference*. Before the modern era of GP approximations, two seminal works distributed the computational load with *local experts* (Jacobs et al., 1991; Hinton, 2002). While the Bayesian committee machine (BCM) of Tresp (2000) focused on merging independently trained GP regression models on data partitions, the infinite mixture of GP experts (Rasmussen and Ghahramani, 2002) combined local GPs to gain expressiveness. Moreover, the emergence of large datasets, with size $N > 10^4$, led to the introduction of variational methods (Titsias, 2009a) for scaling up GP inference. Two recent works that combined sparse GPs with distributed regression ideas are Gal et al. (2014); Deisenroth and Ng (2015). Their approaches are focused in exact GP regression and the distribution of computational load, respectively.

**Contribution.** We present a framework based on modular Gaussian processes. Each module is considered to be a sparse variational GP, containing only variational parameters and hyperparameters. Our main contribution is to keep such *modules* intact, to build meta-GPs without accessing any past observation. This makes the computational cost minimal, becoming a competitor against inference methods for large-datasets. The key principle is the augmentation of integrals, that can be understood in terms of the Kullback-Leibler (KL) divergence between stochastic processes (Matthews et al., 2016). We experimentally provide evidence of the benefits in different applied problems, given several GP architectures. The framework is valid for multi-output settings and heterogeneous likelihoods, that is, handling one data-type per output. Our idea follows the spirit of not being data-driven any longer, but module-driven instead, where the Python user specifies a list of models as input: `models = {model_1, model_2, ..., model_K}`. This will be later called the *dictionary of modules*.

## 2 Modular Gaussian processes

We consider supervised learning problems, where we have an input-output training dataset $\mathcal{D} = \{\boldsymbol{x}_n, y_n\}_{n=1}^N$ with $\boldsymbol{x}_n \in \mathbb{R}^p$. We assume i.i.d. outputs $y_n$, that can be either continuous or discrete variables. For convenience, we will refer to the likelihood term $p(y|\boldsymbol{\theta})$ as $p(y|f)$ where the generative parameters are linked via $\boldsymbol{\theta} = f(\boldsymbol{x})$. We say that $f(\cdot)$ is a non-linear function drawn from a zero-mean GP prior $f \sim \mathcal{GP}(0, k(\cdot, \cdot))$, and $k(\cdot, \cdot)$ is the covariance function or *kernel*. Importantly, when non-Gaussian outputs are considered, the GP output function $f(\cdot)$ might need an extra deterministic mapping $\Phi(\cdot)$ to be transformed to the appropriate parametric domain of $\boldsymbol{\theta}$.

**Data modules.** The dataset $\mathcal{D}$ is assumed to be partitioned into an arbitrary number of $K$ subsets or *modules*, that are observed and processed independently, that is, $\{\mathcal{D}_1, \mathcal{D}_2, \ldots, \mathcal{D}_K\}$. There is not any restriction on the number of modules, $\{\mathcal{D}_k\}_{k=1}^K$ do not need to have the same size, and we only restrict them to be $N_k < N$. However, since we deal with a huge number of observations, we still consider that $N_k$ for all $k \in \{1, 2, \ldots, K\}$ is sufficiently large for not accepting exact GP inference due to temporal and computational demands. Notice that $k$ is an index while $k(\cdot, \cdot)$ is the *kernel*.

### 2.1 Sparse variational approximations for independent modules

For each data module, we adopt the sparse variational GP approach based on *inducing variables*, and the framework of Titsias (2009b), where posterior mean vectors and covariance matrices are computed as parameters. The use of auxiliary variables with approximate inference methods is widely known in the GP literature (Hensman et al., 2013, 2015; Matthews et al., 2016). In the context of $K$ independent data modules, we define subsets of $M_k \ll N_k$ inducing inputs $\boldsymbol{Z}_k = \{\boldsymbol{z}_m\}_{m=1}^{M_k}$, where $\boldsymbol{z}_m \in \mathbb{R}^p$.

Their non-linear function evaluations by $f(\cdot)$ are denoted as $\boldsymbol{u}_k = [f(\boldsymbol{z}_1), f(\boldsymbol{z}_2), \ldots, f(\boldsymbol{z}_{M_k})]^\top$. As a first approach, we assume that $f$ is stationary across all modules, being all $\boldsymbol{u}_k \; \forall k \in \{1, 2, \ldots, K\}$ function evaluations of $f(\cdot)$. This will be later relaxed in the following section.

To obtain the independent approximation to the posterior distribution $p(f|\mathcal{D}_k)$ of the GP, we introduce a $k$th variational distribution $q_k(f)$ for each data module $\mathcal{D}_k$. In particular, this variational density factorises as $q_k(f) = p(f_{\neq \boldsymbol{u}_k}|\boldsymbol{u}_k)q_k(\boldsymbol{u}_k)$, with $q_k(\boldsymbol{u}_k) = \mathcal{N}(\boldsymbol{u}_k|\boldsymbol{\mu}_k, \boldsymbol{S}_k)$ and $p(f_{\neq \boldsymbol{u}_k}|\boldsymbol{u}_k)$ being the standard conditional GP prior distribution given the hyperparameters $\boldsymbol{\psi}_k$ of the $k$th kernel. To fit the GP modules, and their variational distributions $q_k(\boldsymbol{u}_k)$, we build lower bounds $\mathcal{L}_k$ on the marginal log-likelihood (ELBO) of every dataset $\mathcal{D}_k$. Then, we use gradient-based optimisation methods to maximise the $K$ objective functions $\mathcal{L}_k$, one per module, in an separate asynchronous manner. The ELBOs, for either regression or classification tasks are obtained as follows

$$\mathcal{L}_k = \sum_{n=1}^{N_k} \mathbb{E}_{q_k(\mathbf{f}_n)} \left[ \log p(y_n|\mathbf{f}_n) \right] - \mathrm{KL}[q_k(\boldsymbol{u}_k) || p_k(\boldsymbol{u}_k)], \tag{1}$$

with $p_k(\boldsymbol{u}_k) = \mathcal{N}(\boldsymbol{u}_k|\boldsymbol{0}, \mathbf{K}_{kk})$, where $\mathbf{K}_{kk} \in \mathbb{R}^{M_k \times M_k}$ has entries $k(\boldsymbol{z}_m, \boldsymbol{z}_{m'})$ with $\boldsymbol{z}_m, \boldsymbol{z}_{m'} \in \boldsymbol{Z}_k$ and conditioned to certain kernel hyperparameters $\boldsymbol{\psi}_k$ that we also aim to optimise. The variable $\mathbf{f}_n$ corresponds to $f(\boldsymbol{x}_n)$ and the marginal posterior comes from $q_k(\mathbf{f}_n) = \int p(\mathbf{f}_n|\boldsymbol{u}_k)q_k(\boldsymbol{u}_k)d\boldsymbol{u}_k$. In practice, the distributed bounds $\mathcal{L}_k$ are identical to the one presented in Hensman et al. (2015) and can be combined with stochastic variational inference (Hoffman et al., 2013; Hensman et al., 2013).

**Dictionary of modules.** The principal goal here is to obtain a *dictionary*, containing the already fitted GP modules, for their later use. Such dictionary consists, for instance, of a list of module objects $\{\mathcal{M}_1, \mathcal{M}_2, \ldots, \mathcal{M}_K\}$ without any specific order, where each $\mathcal{M}_k = \{\boldsymbol{\phi}_k, \boldsymbol{\psi}_k, \boldsymbol{Z}_k, \mathcal{L}_k^*\}$, being $\boldsymbol{\phi}_k$ the corresponding variational parameters $\boldsymbol{\mu}_k$ and $\boldsymbol{S}_k$. Notice that we also include $\mathcal{L}_k^*$ in addition to the fitted inducing points and hyperparameters. This is the ELBO obtained during optimization. It will play a key role later for the module-driven learning process.

## 2.2   Module-driven lower bound

Ideally, to obtain an inference solution given the GP modules included in the dictionary, the resulting posterior density should be valid for all data modules $\{\mathcal{D}_k\}_{k=1}^K$. This is only possible if we consider the entire dataset $\mathcal{D}$ in a maximum likelihood criterion, that is, using the global log-evidence. Specifically, our goal is now to obtain an approximate posterior distribution $q(f) \approx p(f|\mathcal{D})$, by maximising a lower bound $\mathcal{L}_\mathcal{M}$ under the log-marginal density $\log p(\mathcal{D})$. *This bound should not revisit any data but only the objects in the dictionary of modules*, that are models. Notice that the GP model will be no longer data-driven, but module-driven instead. This is, we build a new model from models, what we call a GP *meta-model* or *meta*-GP.

We begin by considering the full posterior distribution of the stochastic process, similarly as Burt et al. (2019) does for obtaining an upper bound on the Kullback-Leibler (KL) divergence. The idea is to use the large-dimensional integral operators introduced by Matthews et al. (2016) in the context of variational inference, and previously by Seeger (2002) for standard GP error bounds. The use of the large-dimensional integrals is equivalent to an *augment-and-reduce* strategy (Ruiz et al., 2018). It consists of two steps: *i)* we augment the model to be conditioned on the high-dimensional index set of the stochastic process and *ii)* we apply properties of Gaussian marginals to reduce the integral operators to a finite amount of GP function values of interest. Similar strategies have been previously used in continual learning for GPs (Bui et al., 2017; Moreno-Muñoz et al., 2019).

**Global evidence objective.** The construction considered is as follows. We first denote $\boldsymbol{y}$ as all the output targets $\{y_n\}_{n=1}^N$ in the dataset $\mathcal{D}$ and $f_+$ as the *augmented* large-dimensional GP. Notice that $f_+$ sufficiently large and it contains the function values taken by $f(\cdot)$ at both $\{\boldsymbol{x}_n\}_{n=1}^N$ and $\{\boldsymbol{Z}_k\}_{k=1}^K$ for all data modules. We consider it finite. The log-marginal expression is therefore

$$\log p(\boldsymbol{y}) = \log p(\boldsymbol{y}_1, \boldsymbol{y}_2, \ldots, \boldsymbol{y}_K) = \log \int p(\boldsymbol{y}, f_+)df_+, \tag{2}$$

where $\boldsymbol{y}_k = \{y_n\}_{n=1}^{N_k}$ are the output data-points already used for training each GP module $\mathcal{M}_k$. The joint distribution in (2) factorises according to $p(\boldsymbol{y}|f_+)p(f_+)$, where the l.h.s. term is the augmented likelihood distribution and the r.h.s. term would correspond to the GP prior over the large index set of the stochastic process. Then, we introduce the global variational distribution $q(\boldsymbol{u}_*) = \mathcal{N}(\boldsymbol{u}_*|\boldsymbol{\mu}_*, \boldsymbol{S}_*)$

that we aim to fit by maximising a lower bound under $\log p(\boldsymbol{y})$ given the dictionary of modules. The variables $\boldsymbol{u}_*$ correspond to function values of $f(\cdot)$ given the new subset of inducing inputs $\boldsymbol{Z}_* = \{\boldsymbol{z}_m\}_{m=1}^{M}$, where $M$ is the free-complexity degree of the global variational distribution.

To derive the bound, we exploit the reparametrisation introduced by Gal et al. (2014) for distributing the computational load of the expectation term with GP latent variable models (GPLVM) (Lawrence, 2005). This is based on the decoupling of data-points conditioned on the inducing inputs $\boldsymbol{u}_*$. Applying Jensen's inequality, it is obtained as

$$\log p(\boldsymbol{y}) = \log \iint q(\boldsymbol{u}_*) p(f_{+\neq \boldsymbol{u}_*}|\boldsymbol{u}_*) p(\boldsymbol{y}|f_+) \frac{p(\boldsymbol{u}_*)}{q(\boldsymbol{u}_*)} df_{+\neq \boldsymbol{u}_*} d\boldsymbol{u}_*$$

$$\geq \mathbb{E}_{q(\boldsymbol{u}_*)} \left[ \mathbb{E}_{p(\boldsymbol{f}_{+\neq \boldsymbol{u}_*}|\boldsymbol{u}_*)} \left[ \log p(\boldsymbol{y}|f_+) \right] + \log \frac{p(\boldsymbol{u}_*)}{q(\boldsymbol{u}_*)} \right], \quad (3)$$

where we applied the properties of Gaussian conditionals to factorise the GP prior as $p(f_+) = p(\boldsymbol{f}_{+\neq \boldsymbol{u}_*}|\boldsymbol{u}_*)p(\boldsymbol{u}_*)$. Notice that $f_+ = \{\boldsymbol{f}_{+\neq \boldsymbol{u}_*}, \boldsymbol{u}_*\}$. Here, the last prior distribution is $p(\boldsymbol{u}_*) = \mathcal{N}(\boldsymbol{u}_*|\boldsymbol{0}, \mathbf{K}_{**})$ where $[\mathbf{K}_{**}]_{m,n} := k(\boldsymbol{z}_m, \boldsymbol{z}_n)$, with $\boldsymbol{z}_m, \boldsymbol{z}_n \in \boldsymbol{Z}_*$. We also consider $k(\cdot, \cdot)$ conditioned to the global kernel hyperparameters $\psi_*$ that we also aim to estimate. The double expectation in (3) comes from the factorization of the large-dimension integral given $f_+$ and the application of Jensen's inequality twice. Its derivation is in the appendix.

**Likelihood reconstruction.** The augmented likelihood distribution $p(\boldsymbol{y}|f_+)$ is perhaps the most important part of the derivation. It allows us to apply conditional independence (CI) between the data modules $\mathcal{D}_k$, and particularly, the output variables $\boldsymbol{y}_k$. This gives a factorized term that we will later use for introducing the GP modules $\mathcal{M}_k$ stored in our dictionary, that is, $\log p(\boldsymbol{y}|f_+) = \sum_{k=1}^{K} \log p(\boldsymbol{y}_k|f_+)$. To avoid revisiting each $k$th likelihood term, and hence, evaluating the corresponding data which might be unavailable, we want to estimate the likelihood distributions. In particular, we use Bayes theorem conditioned to the large-dimensional augmentation $f_+$. Bayes theorem indicates that each $k$th likelihood can be approximated as

$$p(\boldsymbol{y}_k|f_+) \approx Z_k \frac{q_k(f_+)}{p_k(f_+)} = Z_k \frac{p(\cancel{f_{+\neq \boldsymbol{u}_k}|\boldsymbol{u}_k})q_k(\boldsymbol{u}_k)}{p(\cancel{f_{+\neq \boldsymbol{u}_k}|\boldsymbol{u}_k})p_k(\boldsymbol{u}_k)} = Z_k \frac{q_k(\boldsymbol{u}_k)}{p_k(\boldsymbol{u}_k)}, \quad (4)$$

since $q_k(f_+) \approx p(f_+|\boldsymbol{y}_k)$. Here, $Z_k$ refers to the normalization constant or evidence of $\mathcal{D}_k$, which is still difficult to compute. We also assumed that the augmented approximate density factorises as $q_k(f_+) = p(\boldsymbol{f}_{+\neq \boldsymbol{u}_k}|\boldsymbol{u}_k)q_k(\boldsymbol{u}_k)$, using the structure of the GP prior $p_k(f_+)$, as in the proposition of Titsias (2009a). Similar approximations based on the stochastic process conditionals were previously used in Bui et al. (2017) and Matthews et al. (2016), with emphasis on the theoretical consistency of augmentation. Importantly, all parameters and variables needed for the approximation in (4) are given, and stored in the $k$th module $\mathcal{M}_k$. Then, the nested conditional expectation in (3) turns out to be

$$\mathbb{E}_{p(\boldsymbol{f}_{+\neq \boldsymbol{u}_*}|\boldsymbol{u}_*)} \left[ \log p(\boldsymbol{y}|f_+) \right] \approx \sum_{k=1}^{K} \mathbb{E}_{p(\boldsymbol{u}_k|\boldsymbol{u}_*)} \left[ \log Z_k \frac{q_k(\boldsymbol{u}_k)}{p_k(\boldsymbol{u}_k)} \right],$$

where we applied properties of Gaussian marginals to *reduce* the large-dimensional expectation. For instance, the integral $\int p(f_+)d\boldsymbol{f}_{+\neq \boldsymbol{u}_k}$ is analogous to $\int p(\boldsymbol{f}_{+\neq \boldsymbol{u}_k}, \boldsymbol{u}_k)d\boldsymbol{f}_{+\neq \boldsymbol{u}_k} = p(\boldsymbol{u}_k)$ via marginalisation.

**Variational contrastive expectations.** The introduction of $K$ expectation terms over the log-ratios given the likelihood approximations in (4), leads to particular advantages. Having a *nested* integration in (3), first over $\boldsymbol{u}_*$ given the variational density $q(\boldsymbol{u}_*)$, and second over $\boldsymbol{u}_k$ for the log-ratio $q_k(\boldsymbol{u}_k)/p_k(\boldsymbol{u}_k)$, we can exploit the GP predictive equation to write down

$$\sum_{k=1}^{K} \mathbb{E}_{q(\boldsymbol{u}_*)} \left[ \mathbb{E}_{p(\boldsymbol{u}_k|\boldsymbol{u}_*)} \left[ \log Z_k \frac{q_k(\boldsymbol{u}_k)}{p(\boldsymbol{u}_k)} \right] \right] = \sum_{k=1}^{K} \mathbb{E}_{q_{\mathcal{C}}(\boldsymbol{u}_k)} \left[ \log Z_k \frac{q_k(\boldsymbol{u}_k)}{p_k(\boldsymbol{u}_k)} \right], \quad (5)$$

where we obtained $q_{\mathcal{C}}(\boldsymbol{u}_k)$ via the integral $q_{\mathcal{C}}(\boldsymbol{u}_k) = \int q(\boldsymbol{u}_*)p(\boldsymbol{u}_k|\boldsymbol{u}_*)d\boldsymbol{u}_*$, that coincides with the approximate predictive GP posterior. The computation can be obtained analytically for each $k$th subset $\boldsymbol{u}_k$ using the following expression

$$q_{\mathcal{C}}(\boldsymbol{u}_k) = \mathcal{N}(\boldsymbol{u}_k|\mathbf{K}_{*k}^{\top}\mathbf{K}_{**}^{-1}\boldsymbol{\mu}_*, \mathbf{K}_{kk} + \mathbf{K}_{*k}^{\top}\mathbf{K}_{**}^{-1}(\mathbf{S}_* - \mathbf{K}_{**})\mathbf{K}_{**}^{-1}\mathbf{K}_{*k}),$$

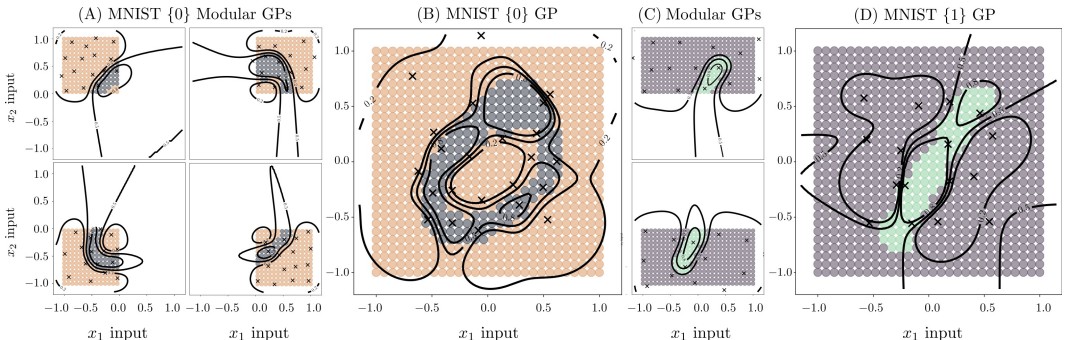

Figure 2: Modular GPs for $\{0, 1\}$ MNIST data samples. The meta-GPs (B–D) are built from fitted modules and do not revisit samples, only parameters and variables stored in each $\mathcal{M}_k$.

where, once again, $\phi_* = \{\mu_*, S_*\}$ are the global variational parameters that we aim to learn. One important detail of the sum of expectations in (5) is that it works as an average contrastive indicator that measures how well the global $q(u_*)$ is being fitted to the modular approximations $q_k(u_k)$. Without the need of revisiting any data, the GP predictive $q_{\mathcal{C}}(u_k)$ is playing a different role in contrast with the usual one. Typically, we assume the approximate posterior fixed and fitted, and we evaluate its performance on some test data points. In this case, it goes in the opposite way, the approximate variational distribution is unfixed, and it is instead evaluated over the inducing inputs $Z_k$ provided by each $k$th module $\mathcal{M}_k$.

**Module-driven lower bound.** We are now able to simplify the initial bound in (3) by substituting the first term with the contrastive expectations presented in (5). This substitution gives us an initial version of the lower bound under the log-marginal likelihood of the GP, where no data intervene,

$$\log p(\boldsymbol{y}) \geq \sum_{k=1}^{K} \log Z_k + \sum_{k=1}^{K} \mathbb{E}_{q_{\mathcal{C}}(\boldsymbol{u}_k)} \left[\log q_k(\boldsymbol{u}_k) - \log p_k(\boldsymbol{u}_k)\right] - \mathrm{KL}\left[q(\boldsymbol{u}_*)||p(\boldsymbol{u}_*)\right].$$

However, since normalization constants $\{Z_k\}_{k=1}^{K}$ are difficult to compute, this immediately implies that the bound is not well defined without them. We therefore, make use of the information provided by the modules. One can show that using the alternative construction of variational lower bounds on sparse GP models, the constant satisfies $\log Z_k = \mathcal{L}_k^* + \mathrm{KL}[q(f, \boldsymbol{u}_k)||p(f, \boldsymbol{u}_k|\boldsymbol{y}_k)]$, where $\mathcal{L}_k^*$ is obtained from Eq. (1). Since we know the KL divergence is greater or equal to zero, and each $\mathcal{L}_k^*$ per module is a lower bound of $\log Z_k$, we have that $\sum_{k=1}^{K} \mathcal{L}_k^* \leq \sum_{k=1}^{K} \log Z_k$. This allows us to write down a closed-form bound $\mathcal{L}_{\mathcal{M}} \leq \log p(\boldsymbol{y})$,

$$\mathcal{L}_{\mathcal{M}} = \sum_{k=1}^{K} \mathcal{L}_k^* + \sum_{k=1}^{K} \mathbb{E}_{q_{\mathcal{C}}(\boldsymbol{u}_k)} \left[\log q_k(\boldsymbol{u}_k) - \log p(\boldsymbol{u}_k)\right] - \mathrm{KL}\left[q(\boldsymbol{u}_*)||p(\boldsymbol{u}_*)\right]. \qquad (6)$$

The maximisation of (6) is w.r.t. the parameters $\phi_*$, the hyperparameters $\psi_*$ and $Z_*$. To assure the positive-definitiness of variational covariance matrices $\{S_k\}_{k=1}^{K}$ and $S_*$ on both local and global cases, we consider that they all factorize according to the Cholesky decomposition $S = LL^\top$. We can then use unconstrained optimization to find optimal values for the lower-triangular matrices $L$.

A priori, the module-driven bound is agnostic with respect to the likelihood model chosen. There is a general derivation in Matthews et al. (2016) of how stochastic processes and their integral operators are affected by projection functions, that is, different linking mappings of the function $f(\cdot)$ to the parameters $\boldsymbol{\theta}$. In such cases, the local lower bounds $\mathcal{L}_k$ in (1) might include expectation terms that are intractable. Since we build the framework to accept any possible data-type, we propose to solve the integrals via Gaussian-Hermite quadratures as in Hensman et al. (2015); Saul et al. (2016), and if this is not possible, an alternative would be to apply Monte-Carlo methods.

## 2.3 Transfer learning with multi-output modules

The arguments used so far are based on the idea that there exists a stationary GP across all modules, being $\boldsymbol{u}_k \ \forall\{1, 2, \ldots, K\}$ values of the same function $f(\cdot)$. This viewpoint can be satisfied if we deal with data from the same domain — see Figure 2 for illustrative examples with MNIST images.

However, regarding the type of outputs that can be modelled, it becomes difficult when we have modules of heterogeneous data from the same phenomena, e.g. a mix of continuous, binary or discrete variables with different likelihood models. In such cases, we know that independent modules in our dictionary $\mathcal{M}$ might be strongly correlated, but not parametrized by one single-output GP. As a result, we relax the stationary assumption over $f$ to accept an additional set of independent latent functions $\mathcal{V} = \{v_q(\cdot)\}_{q=1}^{Q}$. This is, we assume the output functions $\{f_k(\cdot)\}_{k=1}^{K}$ per module $\mathcal{M}_k \in \mathcal{M}$ to be linear combinations of $\mathcal{V}$. Based on Moreno-Muñoz et al. (2018), this is equivalent to introducing a multi-output Gaussian process (MOGP) prior (Alvarez et al., 2012) in the formulation.

**Multi-parameter modules.** Given $Q$ latent functions $\mathcal{V}$, each latent function $v_q(\boldsymbol{x})$ is assumed to be drawn from an independent GP prior, such that $v_q(\cdot) \sim \mathcal{GP}(0, k_q(\cdot, \cdot))$, where $k_q$ can be any valid covariance function, and the zero mean is assumed for simplicity. Since we adopt the linear model of coregionalisation (LMC) (Journel and Huijbregts, 1978), each output function $f_k$ per module $\mathcal{M}_k$ is then given as $f_k(\boldsymbol{x}) = \sum_{q=1}^{Q} \sum_{i=1}^{R_q} a_{k,q}^i v_q^i(\boldsymbol{x})$, where $v_q^i(\boldsymbol{x})$ are i.i.d. samples from each latent $q$th GP prior and $a_{k,q}^i \in \mathbb{R}$ are their associated coefficients for $i = 1, 2, \ldots, R_q$ samples. Notice that the variational parameters per module $\mathcal{M}_k$, are given in terms of $\boldsymbol{u}_k$, the evaluation of $f_k(\cdot)$ given the index set

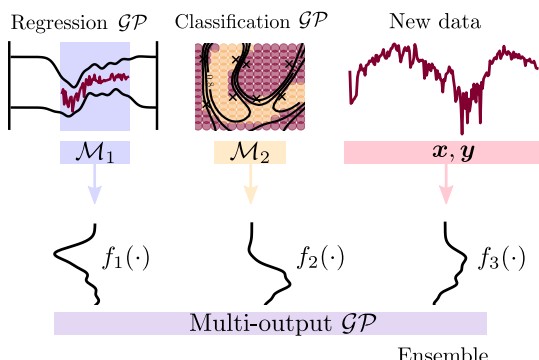

Figure 3: Illustration of modular MOGP. Regression, classification and data tasks are *linked* to outputs $f_k(\cdot)$.

$\boldsymbol{Z}_k$. Notice that the output functions $\{f_k\}_{k=1}^{K}$ are CI, given $\mathcal{V}$. This leads to heterogeneous likelihoods that we can also factorise. The *augmented* distribution becomes $\log p(\boldsymbol{y}|\mathcal{F}_+) = \sum_{k=1}^{K} \log p(\boldsymbol{y}_k|f_{k+})$, where we denote $\mathcal{F}_+ = \{f_{k+}\}_{k=1}^{K}$.

**Augmentation of multiple likelihoods.** We follow the same formalism for latent functions $\mathcal{V}$ used in the single-output scenario, where we introduced the inducing inputs $\boldsymbol{Z}_* = \{\boldsymbol{z}_m\}_{m=1}^{M}$. Sparse approximations in the context of MOGP have been previously explored in Alvarez and Lawrence (2009); Álvarez et al. (2010). Additionally, we define $Q$ variational distributions, such that $q(\boldsymbol{v}_{*q}) = \mathcal{N}(\boldsymbol{v}_{*q}|\boldsymbol{\mu}_{*q}, \boldsymbol{S}_{*q})$, where $\boldsymbol{v}_{*q} = [v_q(\boldsymbol{z}_1), v_q(\boldsymbol{z}_2), \ldots, v_q(\boldsymbol{z}_M)]^\top$. This gives us a way to construct approximations to multi-output likelihood terms $p(\boldsymbol{y}_k|f_{k+})$, and particularly, we use the *augmented* joint distribution $p(\mathcal{F}_+, \mathcal{V}_+)$, that factorises $p(\mathcal{F}_+, \mathcal{V}_+) = p(\mathcal{F}_+, \mathcal{V}_{+\neq*}|\mathcal{V}_*)p(\mathcal{V}_*)$, where $p(\mathcal{V}_*) = \prod_{q=1}^{Q} p(\boldsymbol{v}_{*q})$, GP priors per $q$th latent function are $p(\boldsymbol{v}_{*q}) = \mathcal{N}(0, \mathbf{K}_{*q*q})$ and matrices $[\mathbf{K}_{*q*q}]_{m,n} := k(\boldsymbol{z}_m, \boldsymbol{z}_n)$, with $\boldsymbol{z}_m, \boldsymbol{z}_n \in \boldsymbol{Z}_*$. Since we can obtain closed-form conditionals between latent variables $\boldsymbol{v}_{*q}$ and $\boldsymbol{u}_k$ from modules using the cross-covariance matrices of the MOGP, we are able to rewrite our multi-output module-driven bound as

$$\mathcal{L}_{\mathcal{M}}^{\text{MO}} = \sum_{k=1}^{K} \mathcal{L}_k^* + \sum_{k=1}^{K} \mathbb{E}_{q_\mathcal{C}(\boldsymbol{u}_k)} \left[ \log q_k(\boldsymbol{u}_k) - \log p_k(\boldsymbol{u}_k) \right] - \sum_{q=1}^{Q} \text{KL} \left[ q(\boldsymbol{v}_{*q}) || p(\boldsymbol{v}_{*q}) \right], \quad (7)$$

where we also rewrote the contrastive density $q_\mathcal{C}(\boldsymbol{u}_k) = \int p(\boldsymbol{u}_k|\boldsymbol{v}_*)q(\boldsymbol{v}_*)d\boldsymbol{v}_*$, with $\boldsymbol{v}_* = \{\boldsymbol{v}_{*q}\}_{q=1}^{Q}$ and $q(\boldsymbol{v}_*) = \prod_{q=1}^{Q} q(\boldsymbol{v}_{*q})$. This results in closed-form solution for $q_\mathcal{C}(\boldsymbol{u}_k)$, which is in the appendix.

**Computational cost and connections.** The computational cost of training modules is $\mathcal{O}(N_k M_k^2)$, while the meta-GP reduces to $\mathcal{O}((\sum_k M_k)M^2)$ and $\mathcal{O}(M^2)$ in training and memory, respectively. If we consider the multi-output approach, the cost only increases to $\mathcal{O}((\sum_k M_k)QM^2)$. Note that learning the meta-GP is no longer data-driven. The methods for distributed GPs typically need $\mathcal{O}(\sum_k N_k^2)$ for global prediction. We also find a link between the module-driven bound in (6) and the underlying idea in Tresp (2000); Deisenroth and Ng (2015). To approximate the predictive posterior, such methods combine local estimates divided by the GP prior. This is analogous to (6), where we formulate in the logarithmic plane and the variational inference setup.

Table 1: Properties of distributed/modular GP models

| MODEL | $\mathcal{N}$ REG. | non-$\mathcal{N}$ REG. | CLASS. | HET. | INFERENCE | $\mathcal{GP}_{\text{NODE}}$ | FREE OF DATA ST. |
|---|---|---|---|---|---|---|---|
| Tresp (2000) | ✓ | ✗ | ✗ | ✗ | Analytical | ✓ | ✗ |
| Ng and Deisenroth (2014) | ✓ | ✗ | ✗ | ✗ | Analytical | ✓ | ✗ |
| Cao and Fleet (2014) | ✓ | ✗ | ✗ | ✗ | Analytical | ✓ | ✗ |
| Deisenroth and Ng (2015) | ✓ | ✗ | ✗ | ✗ | Analytical | ✓ | ✗ |
| Gal et al. (2014) | ✓ | ✓ | ✓ | ✗ | Variational | ✗ | ✓ |
| **This work** | ✓ | ✓ | ✓ | ✓ | Variational | ✓ | ✓ |

($^*$) Respectively, Gaussian and non-Gaussian regression ($\mathcal{N}$ & non-$\mathcal{N}$ REG), classification (CLASS), heterogeneous (HET) and storage (ST).

## 3 Related work

In terms of distributed inference for scaling up computation, that is, the delivery of tasks across parallel nodes, we are similar to Gal et al. (2014). However, their approach distributes calculus operations, i.e. product of vectors or matrix inversions that are later centralized, but no usable models under our definition of module. This is, we obtain a meta-model from models that are pre-trained and non-obsolete. Alternatively, if we look to the property of having nodes that contain *usable* GP models (Table 1), we are similar to Deisenroth and Ng (2015); Cao and Fleet (2014); Neumann et al. (2009) and Tresp (2000), with the difference that we introduce variational approximation methods for non-Gaussian likelihoods. An important detail is that the idea of exploiting properties of full stochastic processes (Matthews et al., 2016) for substituting likelihood terms in a general bound has been previously considered in Bui et al. (2017), ending in the derivation of expectation-propagation (EP) methods for streaming inference in GPs. There is also the method of Bui et al. (2018) for both federated and continual learning, but focused both on EP and the Bayesian approach of Nguyen et al. (2018). A short analysis of their application to GPs is included for continual learning settings but far from the large-scale scope of our paper. Moreover, the spirit of using inducing-points as pseudo-approximations of local subsets of data is shared with Bui and Turner (2014), that comments its potential application to distributed setups. More oriented to dynamical modular models, we find the work by Velychko et al. (2018), whose factorisation across tasks is similar to Ng and Deisenroth (2014) but closer to state-space methods. It is also important to mention that we are different from Duvenaud et al. (2013), because we work in the concept of models, not a dictionary of kernels as they do. We also find Mallasto and Feragen (2017), where they incorporate uncertainty in a large population analysis from the optimal transport perspective, using GPs and 2-Wasserstein metrics.

## 4 Experiments

In this section, we evaluate the performance of the module-based GP framework (MODULARGP) for multiple architectures, scenarios and data access settings. To illustrate its usability, we present results in three different learning scenarios: *i)* regression, *ii)* classification and *iii)* multi-output learning where we also consider heterogeneous likelihoods. All experiments are numbered from one to six in roman characters. Performance metrics are given in terms of the negative log-predictive density (NLPD), root mean-square error (RMSE) and mean-absolute error (MAE). For standard optimization, we used the Adam algorithm (Kingma and Ba, 2015). We provide Pytorch code that allows to easily learn the meta models from GP modules.[1] It also includes the baseline methods used. Details about strategies for initialization and optimization are provided in the appendix. We also remark that data are never revisited and their presence in the meta-GP plots is just for clarity.

Table 2: Comparative error metrics for distributed GP models.

| DATA SIZE → | 10K | | | 100K | | | 1M | | |
|---|---|---|---|---|---|---|---|---|---|
| MODEL | NLPD | RMSE | MAE | NLPD | RMSE | MAE | NLPD | RMSE | MAE |
| BCM | $2.99 \pm 0.94$ | $11.94 \pm 18.89$ | $2.05 \pm 1.31$ | $3.51 \pm 0.73$ | $2.33 \pm 0.96$ | $1.34 \pm 1.03$ | NA | NA | NA |
| PoE | $2.79 \pm 0.16$ | $2.32 \pm 0.22$ | $1.86 \pm 0.22$ | $2.82 \pm 0.67$ | $2.19 \pm 0.91$ | $1.71 \pm 0.84$ | $2.91 \pm 0.63$ | $1.98 \pm 0.61$ | $1.32 \pm 0.05$ |
| GPoE | $2.79 \pm 0.56$ | $2.43 \pm 0.52$ | $1.96 \pm 0.48$ | $2.73 \pm 0.72$ | $2.19 \pm 0.91$ | $1.71 \pm 0.84$ | $2.72 \pm 0.52$ | $1.98 \pm 0.61$ | $\mathbf{1.32 \pm 0.05}$ |
| RBCM | $2.96 \pm 0.51$ | $2.49 \pm 0.51$ | $2.02 \pm 0.46$ | $3.03 \pm 0.86$ | $2.51 \pm 1.12$ | $1.99 \pm 1.04$ | $\mathbf{2.56 \pm 0.06}$ | $\mathbf{1.82 \pm 0.02}$ | $1.37 \pm 0.03$ |
| ModularGP | $\mathbf{2.71 \pm 0.11}$ | $\mathbf{1.56 \pm 0.04}$ | $\mathbf{0.97 \pm 0.05}$ | $2.89 \pm 0.07$ | $1.73 \pm 0.01$ | $1.23 \pm 0.02$ | $2.87 \pm 0.09$ | $1.87 \pm 0.07$ | $1.34 \pm 0.09$ |

**Acronyms:** BCM (Tresp, 2000), PoE (Ng and Deisenroth, 2014), GPoE (Cao and Fleet, 2014) and RBCM (Deisenroth and Ng, 2015).

**4.1 Regression.** In our first experiments for sparse variational GP regression, we provide both qualitative and quantitative results about the performance of the modular GP framework.
**(i) Toy concatenation:** In Figure 4, we show three of five tasks united in a new GP model. Tasks are GP modules fitted independently with $N_k$=500 synthetic data points and $M_k$=15 inducing variables

---

[1]The code is publicly available in the repository: `https://github.com/pmorenoz/ModularGP/`.

Table 3: Comparative metrics of modular multi-output GPs for US-FLIGHT dataset.

| PARTITION → | DAYS | | | MONTHS | | |
|---|---|---|---|---|---|---|
| MODELS | NLPD | MAE | RMSE | NLPD | MAE | RMSE |
| MODULES (†) | $2.36 \pm 0.18$ | $1.48 \pm 0.26$ | $2.31 \pm 0.24$ | $2.03 \pm 0.02$ | $1.53 \pm 0.06$ | $1.83 \pm 0.03$ |
| MODULARGP ($Q=2$) | $2.49 \pm 0.37$ | $1.49 \pm 0.26$ | $2.31 \pm 0.24$ | $2.51 \pm 0.34$ | $\mathbf{1.56 \pm 0.14}$ | $2.37 \pm 0.13$ |
| MODULARGP ($Q=3$) | $2.38 \pm 0.23$ | $\mathbf{1.49 \pm 0.25}$ | $2.31 \pm 0.25$ | $2.38 \pm 0.13$ | $1.57 \pm 0.13$ | $2.38 \pm 0.11$ |
| MODULARGP ($Q=4$) | $\mathbf{2.36 \pm 0.15}$ | $1.49 \pm 0.26$ | $2.31 \pm 0.24$ | $2.39 \pm 0.03$ | $1.57 \pm 0.14$ | $2.37 \pm 0.12$ |
| MOGP ($Q=2$) | $2.49 \pm 0.38$ | $1.51 \pm 0.25$ | $2.31 \pm 0.24$ | $2.58 \pm 0.42$ | $1.61 \pm 0.12$ | $2.23 \pm 0.14$ |
| MOGP ($Q=3$) | $2.39 \pm 0.25$ | $1.50 \pm 0.26$ | $2.31 \pm 0.26$ | $2.46 \pm 0.38$ | $1.61 \pm 0.11$ | $2.18 \pm 0.12$ |
| MOGP ($Q=4$) | $2.37 \pm 0.17$ | $1.51 \pm 0.26$ | $2.31 \pm 0.25$ | $\mathbf{2.34 \pm 0.28}$ | $1.63 \pm 0.11$ | $\mathbf{2.14 \pm 0.13}$ |

| PARTITION → | DAYS | | | MONTHS | | |
|---|---|---|---|---|---|---|
| AVG. DIFFERENCE PER OUTPUT → | $\Delta$NLPD | $\Delta$MAE | $\Delta$RMSE | $\Delta$NLPD | $\Delta$MAE | $\Delta$RMSE |
| MODULARGP ($Q=2$) vs. Modules | $-3.91\%$ | $\mathbf{-0.64}\%$ | $-0.17\%$ | $-17.69\%$ | $-1.41\%$ | $-22.73\%$ |
| MODULARGP ($Q=3$) vs. Modules | $-0.51\%$ | $-0.99\%$ | $-0.16\%$ | $-14.19\%$ | $-1.41\%$ | $-22.81\%$ |
| MODULARGP ($Q=4$) vs. Modules | $+\mathbf{0.13}\%$ | $-1,31\%$ | $-0.19\%$ | $-14.93\%$ | $-1.87\%$ | $-22.34\%$ |
| MODULARGP ($Q=2$) vs. MOGP | $-3.71\%$ | $-0.75\%$ | $\mathbf{-0.15}\%$ | $-19.49\%$ | $-4.21\%$ | $-18.04\%$ |
| MODULARGP ($Q=3$) vs. MOGP | $-0.91\%$ | $-1.25\%$ | $-0.33\%$ | $-13.96\%$ | $-3.32\%$ | $-15.66\%$ |
| MODULARGP ($Q=4$) vs. MOGP | $-0.11\%$ | $-1.59\%$ | $-0.27\%$ | $-\mathbf{11.54}\%$ | $-4.92\%$ | $-\mathbf{13.84}\%$ |

(†) Modules as the metric of reference.

per module. The ensemble fits a variational solution of dimension $M=35$. Notice that the variational meta-GP tends to match the uncertainty of the modules.

**(ii) Distributed GPs:** We provide error metrics for the modular GP framework compared with the state-of-the-art models in Table 2. The training data is synthetic and generated as a combination of $\sin(\cdot)$ functions (in the appendix). For the case with 10K observations, we used $K=50$ tasks with $N_k=200$ data-points and $M_k=3$ inducing variables in each GP module. The scenario for 100K is similar but divided into $K=250$ tasks with $N_k=400$. Our method obtains better results than the exact distributed solutions, as the meta-GP finds the average solution among all modular GPs. The baseline methods are based on a combination of solutions, if one is bad-fitted, it has a direct effect on the predictive performance. We also tested the data with the inference setup of Gal et al. (2014), obtaining an NLPD of $2.58\pm0.11$ with 250 nodes for 100K data. It is slightly better than ours and the baseline methods. Note that it only distributes the computation of matrix products and inversions, which is equivalent to the standard model in Hensman et al. (2013).

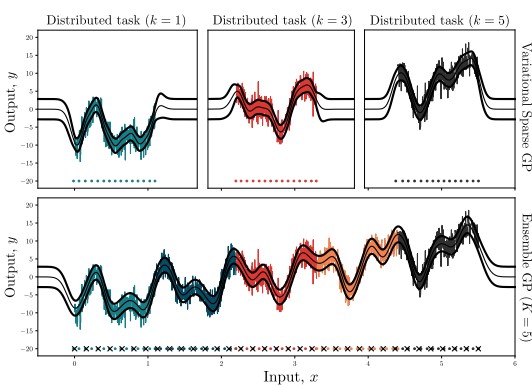

Figure 4: Modular GPs with synthetic data. Three of five tasks are plotted in the upper row.

**(iii) Modular Meta-GPs:** For a large synthetic dataset ($N=10^6$), we tested the modular GP framework with $K=5 \cdot 10^3$ tasks as shown in Table 2. However, if we join large amounts of modular GPs, e.g. $K \gg 10^3$, it is often problematic for baseline methods, due to partitions must be revisited for building final predictions. So data revisiting is not totally avoided. This is not a constraint for us, as we work with the expectation of modules. Table 2 (1M) shows that we are closer to the exact inference solutions, even in large-scale cases. Additionally, we repeated the experiment in a *pyramidal* way — see appendix. That is, building meta-GPs from modular meta-GPs, inspired in Deisenroth and Ng (2015). Our method obtained $\{$NLPD=4.15, RMSE=2.71, MAE=2.27$\}$.

**4.2 Classification.** We adapted the modular GP framework to accept non-Gaussian likelihoods, and in particular, binary classification with Bernoulli distributions. We used the *sigmoid* mapping to link the GP function and the probit parameters. Figure 2 shows an illustrative pixel-wise MNIST $\{0,1\}$ experiment inspired in Van der Wilk et al. (2017), where $\{0,1\}$ meta-GPs are free of data revisiting.

**(iv) Banana dataset:** We used the popular dataset in sparse GP classification for testing our method with $M=25$. We obtained a test NLPD$= 7.21\pm0.04$, while the baseline variational GP test NLPD was $7.29\pm7.85\times10^{-4}$. The improvement is understandable as the total number of inducing points used, including the ones in modules (C), is higher in the modular GP scenario.

**4.3 Multi-output learning.** We tested the performance of multi-output meta-GPs on large datasets. Error metrics are averaged across all output functions $\{f_k\}_{k=1}^K$ and we are interested in the difference

C

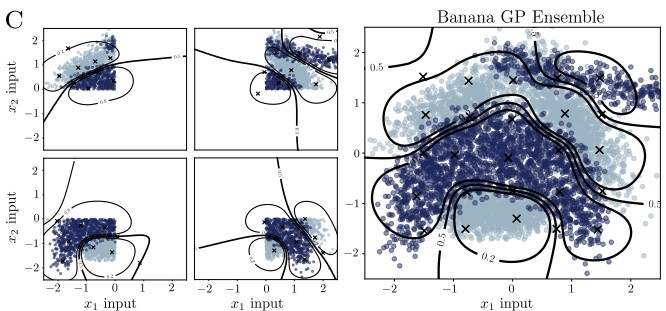
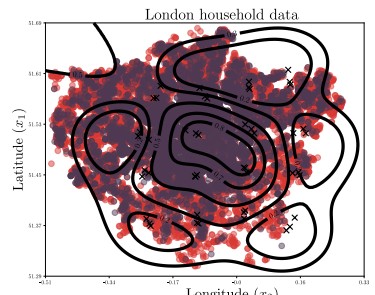

Figure 5: Modular GP classifier for banana dataset.  Figure 6: Modular MOGP $f_2(\cdot)$ .

of predictive performance between the outputs of the meta-GP vs. modules and the standard MOGP model (Alvarez and Lawrence, 2009). Links to datasets are provided in the appendix.

**(v) Airline delays (US flight data):** We took data of US airlines from 2008 (1.5M), where 200K samples where used for testing. The output target is the flight delay and inputs are 6 dimensional. Our goal is to analyse if having GP regression modules per day ($K{=}366$) or per month ($K{=}12$), one can obtain a meta-MOGP linking one-module per output. The results are shown in Table 3, where we see that the relative error difference ($\Delta$) per output is small. This is, the meta-MOGP prediction error only decreases around $1\%$ approx. w.r.t. the GP modules and the variational MOGP baseline.

**(vi) London household:** Based on Hensman et al. (2013), we obtained the register of properties sold in the Greater London County during 2017. The inputs are *longitude-latitude* coordinates and we use an heterogeneous model with an heteroscedastic Gaussian and the two Bernoulli likelihood distributions. We trained GP modules on the Gaussian samples and one of the binary data channels (prices and type of contracts). The scheme is provided in Figure 3. Later, we obtained a meta-MOGP from the two modules and new binary data register (type of house). The predictive curves of the meta-MOGP for the binary module are provided in Figure 6. We obtained NLPD$=4.18{\pm}0.06$ in regression and NLPD$=4.78{\pm}0.03$ in the classification module. The meta-MOGP obtained NLPD$_{f_1}{=}5.49{\pm}0.17$, NLPD$_{f_2}{=}4.64{\pm}0.06$ and NLPD$_{f_3}{=}4.51{\pm}0.11$ in the Gaussian and two Bernoulli tasks respectively. The performance has a slight decrease on the regression task, while improves in the binary output w.r.t. the module. The dataset size is $N{=}20$K. We used $Q{=}3$ and $M{=}Q{\times}16$ inducing inputs.

## 5 Conclusion

We introduced a new framework for building meta-models from independently trained GP modules. Our main contribution is to keep modules intact based on their parameters, avoid their obsolescence and mix them to form new usable tool without revisiting any data. The formulation is principled and allows for GP regression, classification and multi-output learning with heterogeneous likelihoods. We analysed its performance on synthetic and real data, and compared it with the state-of-the-art works. Experimental results show remarkable evidence that the method is robust, and successfully fits to the dictionary of modules. In future work, it would be interesting to extend the framework to include convolutional kernels (Van der Wilk et al., 2017) for large-scale image processing and functional regularisation (Titsias et al., 2020; Moreno-Muñoz et al., 2019) for continual learning applications. As a potential societal impact of our work, we argue that in the long-term, fitted models must be protected, as we do with data to the preserve privacy of individuals.

## Acknowledgements

The authors want to thank Daniel Hernández-Lobato for his constructive comments and Javier González for the useful discussion on this work during the PhD defense of PMM in Madrid last March. We also thank Søren Hauberg for the valuable feedback and the Section for Cognitive Systems at DTU for providing the computational resources. PMM has been supported by FPI grant BES-2016-077626 and ERC funding under the EU's Horizon 2020 research and innovation programme (grant agreement nº 757360). AAR acknowledges the grant TEC2017-92552-EXP (aMBITION) by Ministerio de Ciencia, Innovación y Universidades and the grants TEC2017-86921-C2-2-R (CAIMAN) and RTI2018-099655-B-I00 (CLARA) jointly with the European Comission (ERDF). He also acknowledges the grant Y2018/TCS-4705 (PRACTICO-CM) by the Comunidad de Madrid. MAA has been financed by the EPSRC Research Projects EP/T00343X/2 and EP/V029045/1.

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
