# Modular Gaussian Processes for Transfer Learning
## Supplementary Material

**Pablo Moreno-Muñoz**[*]     **Antonio Artés-Rodríguez**[†]     **Mauricio A. Álvarez**[‡]

[*]Section for Cognitive Systems, Technical University of Denmark (DTU)
[†]Dept. of Signal Theory and Communications, Universidad Carlos III de Madrid, Spain
[†]Evidence-Based Behavior (eB2), Spain
[‡]Dept. of Computer Science, University of Sheffield, UK
pabmo@dtu.dk, antonio@tsc.uc3m.es, mauricio.alvarez@sheffield.ac.uk

In this appendix, we provide additional details about the *modular* GP framework. The full derivation of the evidence lower bound (ELBO) for fitting *meta*-GPs from pre-trained modules is also provided. More insights about the predictive GP posterior used for the contrastive expectation integrals are in the first section. Importantly, we remark that our framework includes a Pytorch tool[1] amenable for multiple testing options and scenarios. Our coding idea is that the Python user only specifies a dictionary of models as input: `models = {model₁, model₂, ..., model_K}`. The code for experiments is included as well as the URLs to the data and the performance metrics used. The *true* generative functions of the toy data and the initial setup of hyperparameters within optimization algorithms are written down at the end of this appendix.

## A    Detailed Derivation of the Module-driven Lower Bound

The construction of module-driven variational bounds from modular GP models is based on the idea of *augmenting* the marginal likelihood to be conditioned on the large-dimensional GP function $f_+$. Notice that $f_+$ contains all the function values taken by $f(\cdot)$ over the input-space $\mathbb{R}^p$, including the input targets $\{\boldsymbol{x}_i\}_{i=1}^N$, the *inducing-inputs* $\{\boldsymbol{Z}_k\}_{k=1}^K$ from modules $\{\mathcal{M}_k\}_{k=1}^K$ and the global ones $\boldsymbol{Z}_*$. Thus, having $K$ partitions of the dataset $\mathcal{D}$ with their corresponding outputs $\boldsymbol{y} = \{\boldsymbol{y}_1, \boldsymbol{y}_2, \ldots, \boldsymbol{y}_K\}$, we begin by augmenting the marginal log-likelihood distribution as

$$\log p(\boldsymbol{y}) = \log p(\boldsymbol{y}_1, \boldsymbol{y}_2, \ldots, \boldsymbol{y}_K) = \log \int p(\boldsymbol{y}, f_+) df_+, \tag{1}$$

that factorises according to

$$\log \int p(\boldsymbol{y}, f_+) df_+ = \log \int p(\boldsymbol{y}|f_+) p(f_+) df_+, \tag{2}$$

where $p(\boldsymbol{y}|f_+)$ is the *augmented* likelihood term of all the output targets of interest and $p(f_+)$ the GP prior over the finite number of points in the input-space $\mathbb{R}^p$. This last distribution takes the form of a multivariate Gaussian, that we avoid to evaluate explicitly in the equations. To build the lower bound on the log-marginal likelihood, we first introduce the global variational distribution $q(\boldsymbol{u}_*) = \mathcal{N}(\boldsymbol{u}_*|\boldsymbol{\mu}_*, \boldsymbol{S}_*)$ into the equation,

$$\log p(\boldsymbol{y}) = \log \int p(\boldsymbol{y}|f_+) p(f_+) df_+ = \log \int \frac{q(\boldsymbol{u}_*)}{q(\boldsymbol{u}_*)} p(\boldsymbol{y}|f_+) p(f_+) df_+$$

$$= \log \iint \frac{q(\boldsymbol{u}_*)}{q(\boldsymbol{u}_*)} p(\boldsymbol{y}|f_+) p(f_{+\neq \boldsymbol{u}_*}|\boldsymbol{u}_*) p(\boldsymbol{u}_*) df_{+\neq \boldsymbol{u}_*} d\boldsymbol{u}_*. \tag{3}$$

---

[1]The code is publicly available in the repository `https://github.com/pmorenoz/ModularGP`.

35th Conference on Neural Information Processing Systems (NeurIPS 2021).

Notice that the differentials $df_+$ have been splitted into $df_{+\neq \boldsymbol{u}_*} d\boldsymbol{u}_*$, and at the same time, we applied properties of Gaussian conditionals in the GP prior to rewrite $p(f_+)$ as $p(f_{+\neq \boldsymbol{u}_*}|\boldsymbol{u}_*)p(\boldsymbol{u}_*)$. When the target variables $\boldsymbol{u}_*$ are explicit in the expression, our second step is the application of the Jensen inequality, similarly as it is done in the reparameterisation of (Gal et al., 2014) for Gaussian process latent variable models (GPLVM), that is

$$
\begin{aligned}
\log p(\boldsymbol{y}) &= \log \iint \frac{q(\boldsymbol{u}_*)}{q(\boldsymbol{u}_*)} p(\boldsymbol{y}|f_+)p(f_{+\neq \boldsymbol{u}_*}|\boldsymbol{u}_*)p(\boldsymbol{u}_*)df_{+\neq \boldsymbol{u}_*}d\boldsymbol{u}_* \\
&= \log \iint q(\boldsymbol{u}_*)p(f_{+\neq \boldsymbol{u}_*}|\boldsymbol{u}_*)p(\boldsymbol{y}|f_+)\frac{p(\boldsymbol{u}_*)}{q(\boldsymbol{u}_*)}df_{+\neq \boldsymbol{u}_*}d\boldsymbol{u}_* \\
&= \log \left( \mathbb{E}_{q(\boldsymbol{u}_*)}\left[ \mathbb{E}_{p(\boldsymbol{f}_{+\neq \boldsymbol{u}_*}|\boldsymbol{u}_*)}\left[ p(\boldsymbol{y}|f_+)\frac{p(\boldsymbol{u}_*)}{q(\boldsymbol{u}_*)} \right] \right] \right) \\
&\geq \mathbb{E}_{q(\boldsymbol{u}_*)}\left[ \log \left( \mathbb{E}_{p(\boldsymbol{f}_{+\neq \boldsymbol{u}_*}|\boldsymbol{u}_*)}\left[ p(\boldsymbol{y}|f_+)\frac{p(\boldsymbol{u}_*)}{q(\boldsymbol{u}_*)} \right] \right) \right] \\
&\geq \mathbb{E}_{q(\boldsymbol{u}_*)}\left[ \mathbb{E}_{p(\boldsymbol{f}_{+\neq \boldsymbol{u}_*}|\boldsymbol{u}_*)}\left[ \log \left( p(\boldsymbol{y}|f_+)\frac{p(\boldsymbol{u}_*)}{q(\boldsymbol{u}_*)} \right) \right] \right] = \mathcal{L}_{\mathcal{M}}. \quad (4)
\end{aligned}
$$

Then, if we have (4), which is the first version of our module-driven lower bound $\mathcal{L}_{\mathcal{M}}$, we can use the augmented likelihood term $p(\boldsymbol{y}|f_+)$ to introduce the variational approximations of modules instead of revisiting the data. This is based on Bayes theorem, and we rewrite

$$
\begin{aligned}
\mathcal{L}_{\mathcal{M}} &= \mathbb{E}_{q(\boldsymbol{u}_*)}\left[ \mathbb{E}_{p(\boldsymbol{f}_{+\neq \boldsymbol{u}_*}|\boldsymbol{u}_*)}\left[ \log p(\boldsymbol{y}|f_+) + \log \left( \frac{p(\boldsymbol{u}_*)}{q(\boldsymbol{u}_*)} \right) \right] \right] \\
&= \mathbb{E}_{q(\boldsymbol{u}_*)}\left[ \mathbb{E}_{p(\boldsymbol{f}_{+\neq \boldsymbol{u}_*}|\boldsymbol{u}_*)}\left[ \log p(\boldsymbol{y}|f_+) \right] - \log \left( \frac{q(\boldsymbol{u}_*)}{p(\boldsymbol{u}_*)} \right) \right] \\
&= \mathbb{E}_{q(\boldsymbol{u}_*)}\left[ \mathbb{E}_{p(\boldsymbol{f}_{+\neq \boldsymbol{u}_*}|\boldsymbol{u}_*)}\left[ \sum_{k=1}^{K} \log p(\boldsymbol{y}_k|f_+) \right] - \log \left( \frac{q(\boldsymbol{u}_*)}{p(\boldsymbol{u}_*)} \right) \right] \\
&= \mathbb{E}_{q(\boldsymbol{u}_*)}\left[ \sum_{k=1}^{K} \mathbb{E}_{p(\boldsymbol{f}_{+\neq \boldsymbol{u}_*}|\boldsymbol{u}_*)}\left[ \log p(\boldsymbol{y}_k|f_+) \right] - \log \left( \frac{q(\boldsymbol{u}_*)}{p(\boldsymbol{u}_*)} \right) \right], \quad (5)
\end{aligned}
$$

where the log-ratio $q(\boldsymbol{u}_*)/p(\boldsymbol{u}_*)$ acts as a constant to the second expectation $\mathbb{E}_{p(\boldsymbol{f}_{+\neq \boldsymbol{u}_*}|\boldsymbol{u}_*)}[\cdot]$ and we applied *conditional independence* (CI) among all the output partitions given the finite latent function values $f_+$. That is, we introduced $p(\boldsymbol{y}|f_+) = \prod_{k=1}^{K} p(\boldsymbol{y}_k|f_+)$ to factorise the expectation term in (5) across the $K$ tasks.

Under the approximation of $p(\boldsymbol{y}_k|f_+)$ obtained by inverting Bayes theorem, we use $p(\boldsymbol{y}_k|f_+) \approx Z_k q_k(f_+)/p_k(f_+)$ to introduce the local posterior distributions $q_k(\cdot)$ and priors $p_k(\cdot)$ in the bound $\mathcal{L}_{\mathcal{M}}$. The variable $Z_k$ is the normalization constant of Bayes theorem. This leads to

$$
\begin{aligned}
\mathcal{L}_{\mathcal{M}} &= \mathbb{E}_{q(\boldsymbol{u}_*)}\left[ \sum_{k=1}^{K} \mathbb{E}_{p(\boldsymbol{f}_{+\neq \boldsymbol{u}_*}|\boldsymbol{u}_*)}\left[ \log p(\boldsymbol{y}_k|f_+) \right] - \log \left( \frac{q(\boldsymbol{u}_*)}{p(\boldsymbol{u}_*)} \right) \right] \\
&\approx \mathbb{E}_{q(\boldsymbol{u}_*)}\left[ \sum_{k=1}^{K} \mathbb{E}_{p(\boldsymbol{f}_{+\neq \boldsymbol{u}_*}|\boldsymbol{u}_*)}\left[ \log \left( Z_k \frac{q_k(f_+)}{p_k(f_+)} \right) \right] - \log \left( \frac{q(\boldsymbol{u}_*)}{p(\boldsymbol{u}_*)} \right) \right] \\
&= \mathbb{E}_{q(\boldsymbol{u}_*)}\left[ \sum_{k=1}^{K} \mathbb{E}_{p(\boldsymbol{f}_{+\neq \boldsymbol{u}_*}|\boldsymbol{u}_*)}\left[ \log \left( Z_k \frac{\cancel{p(\boldsymbol{f}_{+\neq \boldsymbol{u}_k}|\boldsymbol{u}_k)}q_k(\boldsymbol{u}_k)}{\cancel{p(\boldsymbol{f}_{+\neq \boldsymbol{u}_k}|\boldsymbol{u}_k)}p_k(\boldsymbol{u}_k)} \right) \right] - \log \left( \frac{q(\boldsymbol{u}_*)}{p(\boldsymbol{u}_*)} \right) \right] \\
&= \mathbb{E}_{q(\boldsymbol{u}_*)}\left[ \sum_{k=1}^{K} \mathbb{E}_{p(\boldsymbol{f}_{+\neq \boldsymbol{u}_*}|\boldsymbol{u}_*)}\left[ \log \left( Z_k \frac{q_k(\boldsymbol{u}_k)}{p_k(\boldsymbol{u}_k)} \right) \right] - \log \left( \frac{q(\boldsymbol{u}_*)}{p(\boldsymbol{u}_*)} \right) \right], \quad (6)
\end{aligned}
$$

where we now have the explicit distributions $q_k(\boldsymbol{u}_k)$ and $p_k(\boldsymbol{u}_k)$ on the subsets of inducing-inputs $\{\boldsymbol{Z}_k\}_{k=1}^K$. Notice that these ones are stored in the $\mathcal{M}_k$ module. The cancellation of conditionals is a result of the variational factorization (Titsias, 2009). Looking to the last version of the bound in (6), there is still one point that maintains the large-dimensionality, the conditional prior $p(\boldsymbol{f}_{+\neq\boldsymbol{u}_*}|\boldsymbol{u}_*)$ and its corresponding expectation term $\mathbb{E}_{p(\boldsymbol{f}_{+\neq\boldsymbol{u}_*}|\boldsymbol{u}_*)}[\cdot]$. To adapt it to the inducing variables $\boldsymbol{u}_k$ of modules, we apply the following simplification to each $k$-th integral in (6) based in the properties of Gaussian marginals (see section A.1),

$$
\mathbb{E}_{p(\boldsymbol{f}_{+\neq\boldsymbol{u}_*}|\boldsymbol{u}_*)}\left[\log\left(Z_k\frac{q_k(\boldsymbol{u}_k)}{p_k(\boldsymbol{u}_k)}\right)\right] = \int p(\boldsymbol{f}_{+\neq\boldsymbol{u}_*}|\boldsymbol{u}_*)\log\left(Z_k\frac{q_k(\boldsymbol{u}_k)}{p_k(\boldsymbol{u}_k)}\right)d\boldsymbol{f}_{+\neq\boldsymbol{u}_*}
$$

$$
= \iint p(\boldsymbol{f}_{+\neq\{\boldsymbol{u}_*,\boldsymbol{u}_k\}},\boldsymbol{u}_k|\boldsymbol{u}_*)\log\left(Z_k\frac{q_k(\boldsymbol{u}_k)}{p_k(\boldsymbol{u}_k)}\right)d\boldsymbol{f}_{+\neq\{\boldsymbol{u}_*,\boldsymbol{u}_k\}}d\boldsymbol{u}_k
$$

$$
= \int p(\boldsymbol{u}_k|\boldsymbol{u}_*)\log\left(Z_k\frac{q_k(\boldsymbol{u}_k)}{p_k(\boldsymbol{u}_k)}\right)d\boldsymbol{u}_k = \mathbb{E}_{p(\boldsymbol{u}_k|\boldsymbol{u}_*)}\left[\log\left(Z_k\frac{q_k(\boldsymbol{u}_k)}{p_k(\boldsymbol{u}_k)}\right)\right]. \quad (7)
$$

This is the expectation that we plug in the final version of the bound, to obtain

$$
\mathcal{L}_{\mathcal{M}} = \mathbb{E}_{q(\boldsymbol{u}_*)}\left[\sum_{k=1}^K \mathbb{E}_{p(\boldsymbol{u}_k|\boldsymbol{u}_*)}\left[\log\left(Z_k\frac{q_k(\boldsymbol{u}_k)}{p_k(\boldsymbol{u}_k)}\right)\right] - \log\left(\frac{q(\boldsymbol{u}_*)}{p(\boldsymbol{u}_*)}\right)\right]
$$

$$
= \sum_{k=1}^K \mathbb{E}_{q(\boldsymbol{u}_*)}\left[\mathbb{E}_{p(\boldsymbol{u}_k|\boldsymbol{u}_*)}\left[\log\left(Z_k\frac{q_k(\boldsymbol{u}_k)}{p_k(\boldsymbol{u}_k)}\right)\right]\right] - \mathbb{E}_{q(\boldsymbol{u}_*)}\left[\log\left(\frac{q(\boldsymbol{u}_*)}{p(\boldsymbol{u}_*)}\right)\right]
$$

$$
= \sum_{k=1}^K \mathbb{E}_{q(\boldsymbol{u}_*)}\left[\mathbb{E}_{p(\boldsymbol{u}_k|\boldsymbol{u}_*)}\left[\log\left(Z_k\frac{q_k(\boldsymbol{u}_k)}{p_k(\boldsymbol{u}_k)}\right)\right]\right] - \mathrm{KL}\left[q(\boldsymbol{u}_*)||p(\boldsymbol{u}_*)\right]
$$

$$
= \sum_{k=1}^K \log Z_k + \sum_{k=1}^K \mathbb{E}_{q_{\mathcal{C}}(\boldsymbol{u}_k)}\left[\log q_k(\boldsymbol{u}_k) - \log p_k(\boldsymbol{u}_k)\right] - \mathrm{KL}\left[q(\boldsymbol{u}_*)||p(\boldsymbol{u}_*)\right], \quad (8)
$$

where $q_{\mathcal{C}}(\boldsymbol{u}_k)$ is the *contrastive* predictive GP posterior, whose derivation is provided in the section A.2. Importantly, the module-driven bound in (8) includes the $K$ normalization constants of Bayes Theorem. One can show that using the alternative construction of the variational lower bound on sparse GP models, this constant satisfies $\log Z_k = \mathcal{L}_k^* + \mathrm{KL}[q(f,\boldsymbol{u}_k)||p(f,\boldsymbol{u}_k|\boldsymbol{y}_k)]$. Since we know the KL divergence is greater or equal to zero, and each $\mathcal{L}_k^*$ per module is a lower bound of $\log Z_k$, we have that $\sum_{k=1}^K \mathcal{L}_k^* \leq \sum_{k=1}^K \log Z_k$. This allows us to rewrite the bound in (8) as

$$
\mathcal{L}_{\mathcal{M}} = \sum_{k=1}^K \mathcal{L}_k^* + \sum_{k=1}^K \mathbb{E}_{q_{\mathcal{C}}(\boldsymbol{u}_k)}\left[\log q_k(\boldsymbol{u}_k) - \log p_k(\boldsymbol{u}_k)\right] - \mathrm{KL}\left[q(\boldsymbol{u}_*)||p(\boldsymbol{u}_*)\right].
$$

This module-driven bound is the one that we aim to maximise w.r.t. some variational parameters $\phi_*$ and hyperparameters $\psi_*$. For a better comprehension of this point, we provide an extra-view of the bound and the presence of (fixed) local and (unfixed) global parameters in each term. See section A.3. for this.

## A.1  Gaussian marginals for infinite-dimensional integral operators

The properties of Gaussian marginal distributions indicate that, having two *normal*-distributed random variables $\boldsymbol{a}$ and $\boldsymbol{b}$, its joint probability distribution is given by

$$
p(\boldsymbol{a},\boldsymbol{b}) = \mathcal{N}\left(\begin{bmatrix}\boldsymbol{\mu}_a\\\boldsymbol{\mu}_b\end{bmatrix}, \begin{bmatrix}\boldsymbol{\Sigma}_{aa} & \boldsymbol{\Sigma}_{ab}\\\boldsymbol{\Sigma}_{ba} & \boldsymbol{\Sigma}_{bb}\end{bmatrix}\right),
$$

and if we want to marginalize one of that variables out, such as $\int p(\boldsymbol{a},\boldsymbol{b})d\boldsymbol{b}$. It turns to be

$$
\int p(\boldsymbol{a},\boldsymbol{b})d\boldsymbol{b} = p(\boldsymbol{a}) = \mathcal{N}(\boldsymbol{\mu}_a,\boldsymbol{\Sigma}_{aa}).
$$

This same property is applicable to every derivation with GPs. In our case, it is the key point that we use to reduce the large-dimensional integral operators w.r.t. the stochastic process $f_+$. An example

can be found in the expectation $\mathbb{E}_{p(\boldsymbol{f}_{+\neq\boldsymbol{u}_*}|\boldsymbol{u}_*)}\left[\cdot\right]$ of (6). Its final derivation to only integrate on $\boldsymbol{u}_k$ rather than on $\boldsymbol{f}_{+\neq\boldsymbol{u}_*}$ comes from

$$p(\boldsymbol{f}_{+\neq\boldsymbol{u}_*}|\boldsymbol{u}_*) = p(\boldsymbol{f}_{+\neq\{\boldsymbol{u}_*,\boldsymbol{u}_k\}},\boldsymbol{u}_k|\boldsymbol{u}_*)$$

$$= \mathcal{N}\left(\begin{bmatrix}\boldsymbol{m}_{\boldsymbol{f}_{+\neq\{\boldsymbol{u}_*,\boldsymbol{u}_k\}}|\boldsymbol{u}_*}\\\boldsymbol{m}_{\boldsymbol{u}_k|\boldsymbol{u}_*}\end{bmatrix},\begin{bmatrix}\boldsymbol{Q}_{\boldsymbol{f}_{+\neq\{\boldsymbol{u}_*,\boldsymbol{u}_k\}}|\boldsymbol{u}_*} & \boldsymbol{Q}_{\boldsymbol{f}_{+\neq\{\boldsymbol{u}_*,\boldsymbol{u}_k\}},\boldsymbol{u}_k|\boldsymbol{u}_*}\\\boldsymbol{Q}_{\boldsymbol{u}_k,\boldsymbol{f}_{+\neq\{\boldsymbol{u}_*,\boldsymbol{u}_k\}}|\boldsymbol{u}_*} & \boldsymbol{Q}_{\boldsymbol{u}_k|\boldsymbol{u}_*}\end{bmatrix}\right),$$

and if we marginalize over $\boldsymbol{f}_{+\neq\{\boldsymbol{u}_*,\boldsymbol{u}_k\}}|\boldsymbol{u}_*$, ends in the following reduction of the conditional prior expectation

$$\mathbb{E}_{p(\boldsymbol{f}_{+\neq\boldsymbol{u}_*}|\boldsymbol{u}_*)}\left[g(\boldsymbol{u}_k)\right] = \int p(\boldsymbol{f}_{+\neq\boldsymbol{u}_*}|\boldsymbol{u}_*)g(\boldsymbol{u}_k)d\boldsymbol{f}_{+\neq\boldsymbol{u}_*}$$

$$= \iint p(\boldsymbol{f}_{+\neq\{\boldsymbol{u}_*,\boldsymbol{u}_k\}},\boldsymbol{u}_k|\boldsymbol{u}_*)g(\boldsymbol{u}_k)d\boldsymbol{f}_{+\neq\{\boldsymbol{u}_*,\boldsymbol{u}_k\}}d\boldsymbol{u}_k$$

$$= \int p(\boldsymbol{u}_k|\boldsymbol{u}_*)g(\boldsymbol{u}_k)d\boldsymbol{u}_k = \mathbb{E}_{p(\boldsymbol{u}_k|\boldsymbol{u}_*)}\left[g(\boldsymbol{u}_k)\right], \quad (9)$$

where we denote $g(\boldsymbol{u}_k) = \log\left(q_k(\boldsymbol{u}_k)/p_k(\boldsymbol{u}_k)\right)$ and we used

$$\int p(\boldsymbol{f}_{+\neq\{\boldsymbol{u}_*,\boldsymbol{u}_k\}},\boldsymbol{u}_k|\boldsymbol{u}_*)d\boldsymbol{f}_{+\neq\{\boldsymbol{u}_*,\boldsymbol{u}_k\}} = p(\boldsymbol{u}_k) = \mathcal{N}(\boldsymbol{m}_{\boldsymbol{u}_k|\boldsymbol{u}_*},\boldsymbol{Q}_{\boldsymbol{u}_k|\boldsymbol{u}_*}).$$

### A.2 Contrastive predictive GP posterior

The *contrastive* predictive GP posterior distribution $q_{\mathcal{C}}(\boldsymbol{u}_k)$ is obtained from the *nested* integration in (8). We begin its derivation with the l.h.s. expectation term in (8), then

$$\sum_{k=1}^{K}\mathbb{E}_{q(\boldsymbol{u}_*)}\left[\mathbb{E}_{p(\boldsymbol{u}_k|\boldsymbol{u}_*)}\left[\log\left(Z_k\frac{q_k(\boldsymbol{u}_k)}{p_k(\boldsymbol{u}_k)}\right)\right]\right]$$

$$= \sum_{k=1}^{K}\iint q(\boldsymbol{u}_*)p(\boldsymbol{u}_k|\boldsymbol{u}_*)\log\left(Z_k\frac{q_k(\boldsymbol{u}_k)}{p_k(\boldsymbol{u}_k)}\right)d\boldsymbol{u}_kd\boldsymbol{u}_*$$

$$= \sum_{k=1}^{K}\int\underbrace{\left(\int q(\boldsymbol{u}_*)p(\boldsymbol{u}_k|\boldsymbol{u}_*)d\boldsymbol{u}_*\right)}_{q_{\mathcal{C}}(\boldsymbol{u}_k)}\log\left(Z_k\frac{q_k(\boldsymbol{u}_k)}{p_k(\boldsymbol{u}_k)}\right)d\boldsymbol{u}_k, \quad (10)$$

where the conditional GP prior distribution between the module's inducing-inputs $\boldsymbol{u}_k$ and the new ones $\boldsymbol{u}_*$, is $p(\boldsymbol{u}_k|\boldsymbol{u}_*) = \mathcal{N}(\boldsymbol{u}_k|\boldsymbol{m}_{k|*},\boldsymbol{Q}_{k|*})$ with

$$\boldsymbol{m}_{k|*} = \mathbf{K}_{*k}^{\top}\mathbf{K}_{**}^{-1}\boldsymbol{u}_*,$$
$$\boldsymbol{Q}_{k|*} = \mathbf{K}_k - \mathbf{K}_{*k}^{\top}\mathbf{K}_{**}^{-1}\mathbf{K}_{*k},$$

and where covariance matrices are built from $[\mathbf{K}_{**}]_{m,n} := k(\boldsymbol{z}_m,\boldsymbol{z}_n)$ with $\boldsymbol{z}_m,\boldsymbol{z}_n \in \mathbb{R}^p$. Finally, the contrastive predictive GP posterior $q_{\mathcal{C}}(\boldsymbol{u}_k)$ can be computed from the expectation term in (10) as

$$\int q(\boldsymbol{u}_*)p(\boldsymbol{u}_k|\boldsymbol{u}_*)d\boldsymbol{u}_* = q_{\mathcal{C}}(\boldsymbol{u}_k) = \mathcal{N}(\boldsymbol{u}_k|\boldsymbol{m}_{\mathcal{C}},\boldsymbol{S}_{\mathcal{C}}), \quad (11)$$

where the parameters $\boldsymbol{m}_{\mathcal{C}}$ and $\boldsymbol{S}_{\mathcal{C}}$ are

$$\boldsymbol{m}_{\mathcal{C}} = \mathbf{K}_{*k}^{\top}\mathbf{K}_{**}^{-1}\boldsymbol{\mu}_*,$$
$$\boldsymbol{S}_{\mathcal{C}} = \mathbf{K}_k - \mathbf{K}_{*k}^{\top}\mathbf{K}_{**}^{-1}(\boldsymbol{S}_* - \mathbf{K}_{**})\mathbf{K}_{**}^{-1}\mathbf{K}_{*k}.$$

### A.3 Parameters in the module-driven lower bound

We approximate the meta-GP posterior distribution as $q(f,\boldsymbol{u}_*) \approx p(f,\boldsymbol{u}_*|\mathcal{D})$, where we introduce the subset of inducing-inputs $\boldsymbol{Z}_* = \{\boldsymbol{z}_m\}_{m=1}^{M}$ and their corresponding function evaluations, $\boldsymbol{u}_*$. Then, the *explicit* variational distribution given the pseudo-observations $\boldsymbol{u}_*$ is $q(\boldsymbol{u}_*) = \mathcal{N}(\boldsymbol{u}_*|\boldsymbol{\mu}_*,\boldsymbol{S}_*)$.

Previously, we have obtained the *dictionary of modules* $\mathcal{M} = \{\mathcal{M}_1, \mathcal{M}_2, \ldots, \mathcal{M}_K\}$ without any specific order, where each $\mathcal{M}_k = \{\phi_k, \psi_k, \boldsymbol{Z}_k, \mathcal{L}_k^*\}$, $\phi_k$ being the corresponding local variational parameters $\boldsymbol{\mu}_k$ and $\boldsymbol{S}_k$.

If we look to the module-driven lower bound in (8), we omitted the conditioning on both variational parameters and hyperparameters for clarity. However, to make this point clear, we will now rewrite (8) to show the influence of each parameter variable over each term in the final bound. We remark that $\{\phi_k, \psi_k\}_{k=1}^K$ are given and fixed, whilst $\{\phi_*, \psi_*\}$ are the variational parameters and hyperparameters that we aim to fit,

$$\mathcal{L}_{\mathcal{M}}(\phi_*, \psi_*) = \sum_{k=1}^K \mathcal{L}_k^* + \sum_{k=1}^K \mathbb{E}_{q_{\mathcal{C}}(\boldsymbol{u}_k|\phi_*, \psi_*)} \left[ \log q_k(\boldsymbol{u}_k|\phi_k) - \log p_k(\boldsymbol{u}_k|\psi_k) \right]$$
$$- \mathrm{KL}\left[ q(\boldsymbol{u}_*|\phi_*) || p(\boldsymbol{u}_*|\psi_*) \right].$$

We remind that the global variational parameters are $\phi_* = \{\boldsymbol{\mu}_*, \boldsymbol{S}_*\}$, while the hyperparameters would correspond to $\psi_* = \{\ell, \sigma_a\}$ in the case of using the vanilla *kernel*, with $\ell$ being the lengthscale and $\sigma_a$ the amplitude variables. The notation of the modular counterpart is equivalent.

The dependencies of parameters in our Pytorch implementation (`https://github.com/pmorenoz/ModularGP`) are clearly shown and evident from the code structure oriented to objects. It is also amenable for the introduction of new covariance functions and more structured variational approximations if needed.

## A.4 Contrastive predictive GP posterior with multi-output modules

When we consider modular multi-output scenarios, we assume the output functions $\{f_k(\cdot)\}_{k=1}^K$ per module $\mathcal{M}_k \in \mathcal{M}$ to be linear combinations of $Q$ latent functions $\mathcal{V}$. Additionally, we define $Q$ variational distributions, such that $q(\boldsymbol{v}_{*q}) = \mathcal{N}(\boldsymbol{v}_{*q}|\boldsymbol{\mu}_{*q}, \boldsymbol{S}_{*q})$, where $\boldsymbol{v}_{*q} = [v_q(\boldsymbol{z}_1), v_q(\boldsymbol{z}_2), \ldots, v_q(\boldsymbol{z}_M)]^\top$. Similarly as in the single-output case, the exact solution of the contrastive density $q_{\mathcal{C}}(\boldsymbol{u}_k)$ can be obtained from the following derivation, that is

$$q_{\mathcal{C}}(\boldsymbol{u}_k) = \int p(\boldsymbol{u}_k|\boldsymbol{v}_*) q(\boldsymbol{v}_*) d\boldsymbol{v}_*$$
$$= \int \cdots \int p(\boldsymbol{u}_k|\boldsymbol{v}_{*1}, \boldsymbol{v}_{*2}, \ldots, \boldsymbol{v}_{*Q}) \prod_{q=1}^Q q(\boldsymbol{v}_{*q}) d\boldsymbol{v}_{*1} \ldots d\boldsymbol{v}_{*Q} = \mathcal{N}(\boldsymbol{u}_k|\boldsymbol{m}_{\mathcal{C}}, \boldsymbol{S}_{\mathcal{C}}),$$

where the conditional distribution is $p(\boldsymbol{u}_k|\boldsymbol{v}_*) = \mathcal{N}(\boldsymbol{u}_k|\boldsymbol{m}_{\boldsymbol{u}_k|\boldsymbol{v}_*}, \boldsymbol{Q}_{\boldsymbol{u}_k|\boldsymbol{v}_*})$ with

$$\boldsymbol{m}_{\boldsymbol{u}_k|\boldsymbol{v}_*} = \mathbf{K}_{\boldsymbol{u}_k \boldsymbol{v}_*} \mathbf{K}_{\boldsymbol{v}_* \boldsymbol{v}_*}^{-1} \boldsymbol{v}_*,$$
$$\boldsymbol{Q}_{\boldsymbol{u}_k|\boldsymbol{v}_*} = \mathbf{K}_{\boldsymbol{u}_k \boldsymbol{u}_k} - \mathbf{K}_{\boldsymbol{u}_k \boldsymbol{v}_*} \mathbf{K}_{\boldsymbol{v}_* \boldsymbol{v}_*}^{-1} \mathbf{K}_{\boldsymbol{u}_k \boldsymbol{v}_*}^\top.$$

Note that covariance matrices in the latter case are equivalent to the ones shown in section A.2 but considering the *cross-covariance* structure of the multi-output GP prior. Finally, the mean vector $\boldsymbol{m}_{\mathcal{C}}$ and the covariance matrix $\boldsymbol{S}_{\mathcal{C}}$ of the contrastive distribution $q_{\mathcal{C}}(\boldsymbol{u}_k)$ are

$$\boldsymbol{m}_{\mathcal{C}} = \sum_{q=1}^Q \mathbf{K}_{\boldsymbol{u}_k \boldsymbol{v}_{*q}} \mathbf{K}_{\boldsymbol{v}_{*q} \boldsymbol{v}_{*q}}^{-1} \boldsymbol{\mu}_{*q},$$
$$\boldsymbol{S}_{\mathcal{C}} = \mathbf{K}_{\boldsymbol{u}_k \boldsymbol{u}_k} + \sum_{q=1}^Q \mathbf{K}_{\boldsymbol{u}_k \boldsymbol{v}_{*q}} \mathbf{K}_{\boldsymbol{v}_{*q} \boldsymbol{v}_{*q}}^{-1} (\boldsymbol{S}_{*q} - \mathbf{K}_{\boldsymbol{v}_{*q} \boldsymbol{v}_{*q}}) \mathbf{K}_{\boldsymbol{v}_{*q} \boldsymbol{v}_{*q}}^{-1} \mathbf{K}_{\boldsymbol{u}_k \boldsymbol{v}_{*q}}^\top.$$

## B Distributions and Expectations

To assure the future and easy reproducibility of our modular GP framework, we provide the exact expression of all distributions and expectations involved in the module-driven lower bound in (8).

**Distributions:** The log-distributions and distributions that appear in (8) are $\log q(\boldsymbol{u}_k)$, $\log p(\boldsymbol{u}_k)$, $q(\boldsymbol{u}_*)$, $p(\boldsymbol{u}_*)$ and $q_{\mathcal{C}}(\boldsymbol{u}_k)$. First, the computation of the logarithmic distributions is

$$\log q(\boldsymbol{u}_k) = \log\left(\mathcal{N}(\boldsymbol{u}_k|\boldsymbol{\mu}_k, \boldsymbol{S}_k)\right) = -\frac{1}{2}(\boldsymbol{u}_k - \boldsymbol{\mu}_k)^\top \boldsymbol{S}_k^{-1}(\boldsymbol{u}_k - \boldsymbol{\mu}_k) - \frac{1}{2}\log \det(2\pi \boldsymbol{S}_k),$$

$$\log p(\boldsymbol{u}_k) = \log\left(\mathcal{N}(\boldsymbol{u}_k|\boldsymbol{0},\mathbf{K}_{kk})\right) = -\frac{1}{2}\boldsymbol{u}_k^\top \mathbf{K}_{kk}^{-1}\boldsymbol{u}_k - \frac{1}{2}\log\det(2\pi\mathbf{K}_{kk}),$$

while $q(\boldsymbol{u}_*)$ and $p(\boldsymbol{u}_*)$ are just $q(\boldsymbol{u}_*) = \mathcal{N}(\boldsymbol{u}_*|\boldsymbol{\mu}_*,\boldsymbol{S}_*)$ and $p(\boldsymbol{u}_*) = \mathcal{N}(\boldsymbol{u}_*|\boldsymbol{0},\mathbf{K}_{**})$. The exact expression of the distribution $q_\mathcal{C}(\boldsymbol{u}_k)$ is provided in the section A.2.

**Expectations and divergences:** The $K$ expectations in the l.h.s. term in (8) can be rewritten as

$$\sum_{k=1}^K \mathbb{E}_{q_\mathcal{C}(\boldsymbol{u}_k)}\left[\log q_k(\boldsymbol{u}_k) - \log p_k(\boldsymbol{u}_k)\right]$$

$$= \sum_{k=1}^K \left[\mathbb{E}_{q_\mathcal{C}(\boldsymbol{u}_k)}\left[\log q_k(\boldsymbol{u}_k)\right] - \mathbb{E}_{q_\mathcal{C}(\boldsymbol{u}_k)}\left[\log p_k(\boldsymbol{u}_k)\right]\right]$$

$$= \sum_{k=1}^K \left[\Big\langle \log q_k(\boldsymbol{u}_k)\Big\rangle_{q_\mathcal{C}(\boldsymbol{u}_k)} - \Big\langle \log p_k(\boldsymbol{u}_k)\Big\rangle_{q_\mathcal{C}(\boldsymbol{u}_k)}\right], \quad (12)$$

where the $k$-th expectations over both $\log q_k(\boldsymbol{u}_k)$ and $\log p_k(\boldsymbol{u}_k)$ take the form

$$\Big\langle \log q_k(\boldsymbol{u}_k)\Big\rangle_{q_\mathcal{C}(\boldsymbol{u}_k)} = -\frac{1}{2}\left(\mathrm{Tr}\left(\boldsymbol{S}_k^{-1}\boldsymbol{S}_\mathcal{C}\right) + (\boldsymbol{m}_\mathcal{C} - \boldsymbol{\mu}_k)^\top \boldsymbol{S}_k^{-1}(\boldsymbol{m}_\mathcal{C} - \boldsymbol{\mu}_k) + \log\det\left(2\pi\boldsymbol{S}_k\right)\right),$$

$$\Big\langle \log p_k(\boldsymbol{u}_k)\Big\rangle_{q_\mathcal{C}(\boldsymbol{u}_k)} = -\frac{1}{2}\left(\mathrm{Tr}\left(\mathbf{K}_{kk}^{-1}\boldsymbol{S}_\mathcal{C}\right) + \boldsymbol{m}_\mathcal{C}^\top \mathbf{K}_{kk}^{-1}\boldsymbol{m}_\mathcal{C} + \log\det\left(2\pi\mathbf{K}_{kk}\right)\right).$$

## C  Combined Module-driven Bounds with New Unseen Data

There might be scenarios where it could be not necessary to distribute the whole dataset $\mathcal{D}$ in $K$ local tasks or, for instance, a new *unseen* subset $k+1$ of observations might be available for processing. In such case, it is still possible to obtain a *combined* meta-GP solution that fits both to the GP module approximations and the new data. For clarity on this point, we rewrite the principal steps of the lower bound derivation in section A but without substituting all the log-likelihood terms by their Bayesian approximation, that is

$$\mathcal{L}_\mathcal{M} = \mathbb{E}_{q(\boldsymbol{u}_*)}\left[\mathbb{E}_{p(\boldsymbol{f}_{+\neq \boldsymbol{u}_*}|\boldsymbol{u}_*)}\left[\sum_{k=1}^K \log(Z_k p(\boldsymbol{y}_k|f_+)) + \log p(\boldsymbol{y}_{k+1}|f_+)\right] - \log\left(\frac{q(\boldsymbol{u}_*)}{p(\boldsymbol{u}_*)}\right)\right]$$

$$= \mathbb{E}_{q(\boldsymbol{u}_*)}\left[\sum_{k=1}^K \mathbb{E}_{p(\boldsymbol{f}_{+\neq \boldsymbol{u}_*}|\boldsymbol{u}_*)}\left[\log\left(Z_k p(\boldsymbol{y}_k|f_+)\right)\right] + \mathbb{E}_{p(\boldsymbol{f}_{+\neq \boldsymbol{u}_*}|\boldsymbol{u}_*)}\left[\log p(\boldsymbol{y}_{k+1}|f_+)\right]\right]$$

$$- \mathbb{E}_{q(\boldsymbol{u}_*)}\left[\log\left(\frac{q(\boldsymbol{u}_*)}{p(\boldsymbol{u}_*)}\right)\right]$$

$$= \mathbb{E}_{q(\boldsymbol{u}_*)}\left[\sum_{k=1}^K \mathbb{E}_{p(\boldsymbol{u}_k|\boldsymbol{u}_*)}\left[\log\left(Z_k \frac{q_k(\boldsymbol{u}_k)}{p_k(\boldsymbol{u}_k)}\right)\right] + \mathbb{E}_{p(\boldsymbol{f}_{k+1}|\boldsymbol{u}_*)}\left[\log p(\boldsymbol{y}_{k+1}|\boldsymbol{f}_{k+1})\right]\right]$$

$$- \mathbb{E}_{q(\boldsymbol{u}_*)}\left[\log\left(\frac{q(\boldsymbol{u}_*)}{p(\boldsymbol{u}_*)}\right)\right]$$

$$= \sum_{k=1}^K \log Z_k + \sum_{k=1}^K \mathbb{E}_{q_\mathcal{C}(\boldsymbol{u}_k)}\left[\log q_k(\boldsymbol{u}_k) - \log p_k(\boldsymbol{u}_k)\right] + \sum_{n=1}^{N_{k+1}} \mathbb{E}_{q(\boldsymbol{f}_n)}\left[\log p(y_n|\boldsymbol{f}_n)\right]$$

$$- \mathrm{KL}\left[q(\boldsymbol{u}_*)\|p(\boldsymbol{u}_*)\right], \quad (13)$$

where $q(\boldsymbol{f}_n)$ is the result of the integral $q(\boldsymbol{f}_n) = \int q(\boldsymbol{u}_*)p(\boldsymbol{f}_n|\boldsymbol{u}_*)d\boldsymbol{u}_*$ and we applied the factorisation to the *new* $(k+1)$-th expectation term as in Hensman et al. (2015).

## D  Detailed Derivation of Multi-output Module-driven Lower Bounds

The first decision that we take for introducing the multi-output GP setting into the modular framework is to relax the assumption of a single latent GP function $f(\cdot)$ modulating all the independent modules.

Instead, we assume them to be correlated and accept a new parameterization based on a set of $Q$ latent functions $\mathcal{V} = \{v_q(\cdot)\}_{q=1}^Q$. Thus, each one of the parameter functions $\{f_k(\cdot)\}_{k=1}^K$ inferred for every GP module $\mathcal{M}_k \in \mathcal{M}$ is assumed to be a linear combination of $\mathcal{V}$.

Having $K$ partitions of the dataset $\mathcal{D}$ with their corresponding outputs $\boldsymbol{y} = \{\boldsymbol{y}_1, \boldsymbol{y}_2, \ldots, \boldsymbol{y}_K\}$ as in Sec. A in this appendix, we begin by augmenting the marginal log-likelihood distribution as

$$\log p(\boldsymbol{y}) = \log p(\boldsymbol{y}_1, \boldsymbol{y}_2, \ldots, \boldsymbol{y}_K) = \log \iint p(\boldsymbol{y}, \mathcal{F}_+, \mathcal{V}_+) d\mathcal{F}_+ d\mathcal{V}_+, \tag{14}$$

where we have denoted $\mathcal{F}_+ = \{f_{k+}\}_{k=1}^K$ and $\mathcal{V}_+ = \{v_{q+}(\cdot)\}_{q=1}^Q$. Importantly, notice that each $f_{k+}$ is the *augmented* large-dimensional GP per $k$th module. This is different from the usual $f_+$ in the single-output formulation, as we now assume that modules are parametrized by correlated functions $f_k(\cdot)$. Then, the log-marginal likelihood factorises as

$$\log \iint p(\boldsymbol{y}, \mathcal{F}_+, \mathcal{V}_+) d\mathcal{F}_+ d\mathcal{V}_+ = \log \iint p(\boldsymbol{y}|\mathcal{F}_+) p(\mathcal{F}_+, \mathcal{V}_+) d\mathcal{F}_+ d\mathcal{V}_+, \tag{15}$$

where the *augmented* likelihood distribution factorises as $p(\boldsymbol{y}|\mathcal{F}_+) = \prod_{k=1}^K p(\boldsymbol{y}_k|f_{k+})$. To build the lower bound under the log-marginal likelihood, we first introduce $Q$ variational distributions such that $q(\boldsymbol{v}_{*q}) = \mathcal{N}(\boldsymbol{v}_{*q}|\boldsymbol{\mu}_{*q}, \boldsymbol{S}_{*q})$, where $\boldsymbol{v}_{*q} = [v_q(\boldsymbol{z}_1), v_q(\boldsymbol{z}_2), \ldots, v_q(\boldsymbol{z}_M)]^\top$. Then,

$$\log p(\boldsymbol{y}) = \log \iint p(\boldsymbol{y}|\mathcal{F}_+) p(\mathcal{F}_+, \mathcal{V}_+) d\mathcal{F}_+ d\mathcal{V}_+ = \log \iint \frac{q(\mathcal{V}_*)}{q(\mathcal{V}_*)} p(\boldsymbol{y}|\mathcal{F}_+) p(\mathcal{F}_+, \mathcal{V}_+) d\mathcal{F}_+ d\mathcal{V}_+$$

$$= \log \iint \frac{q(\mathcal{V}_*)}{q(\mathcal{V}_*)} p(\boldsymbol{y}|\mathcal{F}_+) p(\mathcal{F}_+, \mathcal{V}_{+\neq *}|\mathcal{V}_*) p(\mathcal{V}_*) d\mathcal{F}_+ d\mathcal{V}_{+\neq *} d\mathcal{V}_*, \tag{16}$$

where $q(\mathcal{V}_*) = \prod_{q=1}^Q q(\boldsymbol{v}_{*q})$, $p(\mathcal{V}_*) = \prod_{q=1}^Q p(\boldsymbol{v}_{*q})$ and we used $\mathcal{V}_+ = \{\mathcal{V}_{+\neq *}, \mathcal{V}_*\}$. We also remark the GP priors per $q$th latent function are $p(\boldsymbol{v}_{*q}) = \mathcal{N}(0, \mathbf{K}_{*q*q})$ and matrices $[\mathbf{K}_{*q*q}]_{m,n} := k(\boldsymbol{z}_m, \boldsymbol{z}_n)$, with $\boldsymbol{z}_m, \boldsymbol{z}_n \in \boldsymbol{Z}_*$. Similarly as in the single-output case and derivation included in Sec. A, this log-marginal likelihood distribution can be lower bounded as

$$\log p(\boldsymbol{y}) = \log \iint \frac{q(\mathcal{V}_*)}{q(\mathcal{V}_*)} p(\boldsymbol{y}|\mathcal{F}_+) p(\mathcal{F}_+, \mathcal{V}_{+\neq *}|\mathcal{V}_*) p(\mathcal{V}_*) d\mathcal{F}_+ d\mathcal{V}_{+\neq *} d\mathcal{V}_*$$

$$= \log \iint q(\mathcal{V}_*) p(\boldsymbol{y}|\mathcal{F}_+) p(\mathcal{F}_+, \mathcal{V}_{+\neq *}|\mathcal{V}_*) \frac{p(\mathcal{V}_*)}{q(\mathcal{V}_*)} d\mathcal{F}_+ d\mathcal{V}_{+\neq *} d\mathcal{V}_*$$

$$= \log \left( \mathbb{E}_{q(\mathcal{V}_*)} \left[ \mathbb{E}_{p(\mathcal{F}_+, \mathcal{V}_{+\neq *}|\mathcal{V}_*)} \left[ p(\boldsymbol{y}|\mathcal{F}_+) \frac{p(\mathcal{V}_*)}{q(\mathcal{V}_*)} \right] \right] \right)$$

$$\geq \left( \mathbb{E}_{q(\mathcal{V}_*)} \left[ \mathbb{E}_{p(\mathcal{F}_+, \mathcal{V}_{+\neq *}|\mathcal{V}_*)} \left[ \log p(\boldsymbol{y}|\mathcal{F}_+) \frac{p(\mathcal{V}_*)}{q(\mathcal{V}_*)} \right] \right] \right) = \mathcal{L}_{\mathcal{M}}^{\text{MO}}. \tag{17}$$

This multi-output module-driven bound $\mathcal{L}_{\mathcal{M}}^{\text{MO}}$ is the one included in Eq. 7 in the main manuscript. The derivation of integrals is analogous to the single-output case, considering $Q$ independent variational distributions instead.

## E   Intractable Expectations

When we consider a binary classification task, the likelihood function use to be a Bernoulli distribution, such as $p(y_n|\boldsymbol{f}_n) = \text{Ber}(y_n|\rho = \phi(\boldsymbol{f}_n))$. The non-linear linking mapping $\phi(\cdot)$ is the *sigmoid* function in our case. However, for training the modular GP approximations, the expectation term of the ELBO is still intractable over the log-likelihood distribution. To solve the following integrals

$$\mathbb{E}_{q(\boldsymbol{f}_n)} [\log p(y_n|\boldsymbol{f}_n)] = \int q(\boldsymbol{f}_n) \log p(y_n|\boldsymbol{f}_n) d\boldsymbol{f}_n,$$

we make use of the Gaussian-Hermite quadratures. In the univariate case with binary observations, the previous integral can be approximated as

$$\mathbb{E}_{q(\boldsymbol{f}_n)}\left[\log p(y_n|\boldsymbol{f}_n)\right] \approx \frac{1}{\sqrt{\pi}}\sum_{s=1}^{S} w_s \log p(y_n|\sqrt{2\boldsymbol{v}_n}\boldsymbol{f}_s + \boldsymbol{m}_n),$$

where $\boldsymbol{m}_n$ and $\boldsymbol{v}_n$ are the corresponding mean and variance of the marginal variational distribution $q(\boldsymbol{f}_n)$. Additionally, the pairs of weight-point values $(w_s, \boldsymbol{f}_s)$ are obtained by sampling $S$ times the Hermite polynomial $H_n(x) = (-1)^n e^{x^2}\frac{d^n}{dx^n}e^{-x^2}$. This computation is also used for the calculus of predictive distributions and NLPD metrics.

## F  Experiments, Optimization Algorithms and Metrics

The code for the experiments is written in Python 3.7 and uses the Pytorch syntax for the automatic differentiation of the probabilistic models. It can be found in the repository `https://github.com/pmorenoz/ModularGP`, where we also use the library GPy for some algebraic utilities. In this section, we provide a detailed description of the experiments and the data used, the initialization of both variational parameters and hyperparameters, the optimization algorithm for both the local and the global GP and the performance metrics included in the main manuscript, e.g. the negative log-predictive density (NLPD), the root mean square error (RMSE) and the mean absolute error (MAE).

### F.1  Detailed description of experiments

In our experiments with toy data, we used two versions of the same sinusoidal function, one of them with an incremental bias. The true expressions of $f(\cdot)$ are

$$f(x) = \frac{9}{2}\cos\left(2\pi x + \frac{3\pi}{2}\right) - 3\sin\left(\frac{43\pi}{10}x + \frac{3\pi}{10}\right),$$

and

$$f(x)_{\text{bias}} = f(x) + 3x - \frac{15}{2}.$$

**i) Toy concatenation:** For the first experiment, whose results are illustrated in the Figure 2 of the main manuscript, we generated $K = 5$ subsets of observations in the input-space range $\boldsymbol{x} \in [0.0, 5.5]$. Each subset was formed by $N_k = 500$ uniform samples of $\boldsymbol{x}_k$ that were later evaluated by $f(x)_{\text{bias}}$. Having the values of the true underlying function $\boldsymbol{f}_k = f(\boldsymbol{x}_k)$, we generated the true output targets as $\boldsymbol{y}_k = \boldsymbol{f}_k + \epsilon_k$, where $\epsilon_k \sim \mathcal{N}(0, 2)$. For each modular task, we set a number of $M_k = 15$ inducing-inputs $\boldsymbol{Z}_k$ that were initially equally spaced in each input region. The chosen number of inducing-inputs $\boldsymbol{Z}_*$ for the meta-GP was $M = 35$, initialized in the same manner as in the modular GP case. For all the predictive posterior distributions plotted, we used $N_{\text{test}} = 400$ also equally spaced in the global input-space. The setup of the VEM algorithm (see section F.3) was $\{\text{VE} = 30, \text{VM} = 10, \eta_m = 10^{-3}, \eta_L = 10^{-6}, \eta_\psi = 10^{-8}, \eta_Z = 10^{-8}\}$ for the meta-GP. The previous variables $\eta$ and VM refer to the learning rates used for each type of parameter and the number of iterations in the optimization algorithm.

**ii) Distributed GPs:** In this second experiment, our goal is to compare the performance of the modular GP framework with the distributed GP methods in the literature (Tresp, 2000; Ng and Deisenroth, 2014; Cao and Fleet, 2014; Deisenroth and Ng, 2015). To do so, we begin by generating toy samples from the sinusoidal function $f(x)$. The comparative experiment is divided in two parts, in one, we observe $N = 10^3$ and in the other, $N = 10^4$ input-output data points. In the first case, we splitted the dataset $\mathcal{D}$ in $K = 50$ tasks with $N_k = 200$ and $M_k = 3$ per partition. Any of these distributed subsets were overlapping, and their corresponding input-spaces concatenated perfectly in the range $\boldsymbol{x} \in [0.0, 5.5]$. For the setting with $N = 10^4$ samples, we used $K = 500$ local tasks, that in this case, were overlapping. As we already commented in the main manuscript, the baseline methods underperform more than our framework in problems where partitions do not overlap in the input-space. Additionally, *standard deviation* (std.) values in Table 3 indicate that we are more robust to the fitting crash of some task. This fact is understandable as our method searches a global solution $q(\boldsymbol{u}_*)$ that fits to all the GP modules in average. In contrast, the baseline methods are based on a final ensemble solution that is an analytical combination of all the distributed ones. Then, if one or more fails, the final predictive performance might be catastrophic. Notice that the baseline methods only require to train the GP modules separately, thing that we did with the LBFGS optimization algorithm. The setup of the VEM algorithm during the ensemble fitting was $\{\text{VE} = 30, \text{VM} = 10, \eta_m = 10^{-3},$

$\eta_L = 10^{-6}, \eta_\psi = 10^{-8}, \eta_Z = 10^{-8}\}$. As in the previous experiment with toy data, we set $M = 35$ inducing-inputs.

**iii) Modular Meta-GPs:** For simulating potential scenarios with at least $N = 10^6$ input-output data points, we used the setting of the previous experiment, but with $K = 5 \cdot 10^3$ tasks of $N_k = 800$ instead. However, as explained in the paper, its performance was hard to evaluate in the baseline methods, due to the problem of combining bad-fitted GP models. Then, based on the experiments of Deisenroth and Ng (2015) and the idea of building meta-GPs of meta-GPs, we set a *pyramidal* way for joining the distributed GP modules. It was formed by two *layers*, that is, we joined ensembles twice as shown in the Figure 1 of this appendix.

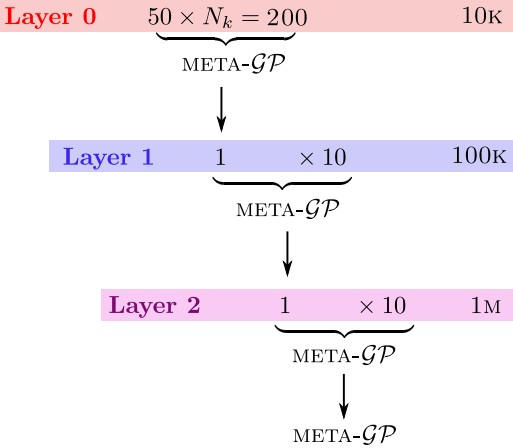

Figure 1: Graphical depiction of the *pyramidal* structure for meta-GP models of meta-GPs.

**iv) Banana dataset:** The banana experiment is perhaps one of the most used datasets for testing GP classification models. We followed a similar strategy as the one used in the illustrative MNIST experiment. After removing the $33\%$ of samples for testing, we partitioned the input-area in four quadrants, i.e. as is shown in Figure 5. For each partition we set a grid of $M_k = 9$ inducing-inputs and later, the maximum complexity of the meta-GP model was set to $M = 25$. The baseline GP classification method also used $M = 25$ inducing-inputs and obtained an NLPD value of $7.29 \pm 7.85 \times 10^{-4}$ after ten trials with different initializations. Our method obtained a test NLPD of $7.21 \pm 0.04$. As we mentioned in the main manuscript, the difference is understandable as the recyclable GP framework used a total amount of $4 \times 16$ inducing-inputs, that capture more uncertainty than the 16 of the baseline method. The setup of the VEM algorithm was $\{\text{VE} = 20, \text{VM} = 10, \eta_m = 10^{-3}, \eta_L = 10^{-5}, \eta_\psi = 10^{-6}, \eta_Z = 10^{-5}\}$.

**v) Airline delays (US flight data):** Based on the experiments presented in Gal et al. (2014); Deisenroth and Ng (2015); Hensman et al. (2013, 2015), we obtained a large-scale dataset reporting the delays of flights in US during the year 2008. The data is publicly available at `https://community.amstat.org/jointscsg-section/dataexpo/dataexpo2009`. The collection is huge, with approximately $N = 1.5 \times 10^6$ input-output data points. Similarly as in the experiments of Deisenroth and Ng (2015), we considered the input space to be $6-$dimensional. The input features correspond to: *i)* year, *ii)* month, *iii)* day, *iv)* day-of-week, *v)* time of departure and *vi)* distance of flight. The output target is the *delay* of the flight in minutes, if there is no delay, the target value is $0$. The pre-processing script for the experiments is also included in the repository. For the experiments we used a total number of $M = 100$ inducing points $\mathbf{Z}_*$. The meta-GP optimization process was carried out using the Adam optimizer (Kingma and Ba, 2015). The setup of learning rates was the following: $\{\eta_m = 10^{-2}, \eta_L = 10^{-3}, \eta_\psi = 10^{-5}, \eta_Z = 10^{-5}\}$. We used the ARD version of the *vanilla* kernel for this experiment.

**vi) London household data:** Based on the large scale experiments in Hensman et al. (2013), we obtained the register of properties sold in the Greater London county during the 2017 year (`https://www.gov.uk/government/collections/price-paid-data`). All addresses of household registers were translated to *latitude-longitude* coordinates, that we used as the input data points. In our experiment, we selected three heterogeneous registers, one real-valued and the two others are binary. The real-valued output targets correspond to the log-price of the properties included in the registers. Moreover, the binary values make reference to the type of contract, $y_i = 1$ if it was a *leasehold* and $y_i = 0$ if *freehold*, and to the type of property, $y_i = 1$ if it was a *flat*-type and $y_i = 0$ otherwise. Interestingly, we appreciated that both tasks share modes across the input region, as they are correlated. That is, if there is more presence of some type of contract, it makes sense that the price increases or decreases accordingly.

**vii) Pixel-wise MNIST classification:** For the illustrative experiment, we took images of *ones* and *zeros* from the MNIST dataset. To simulate a pixel-wise unsupervised classification problem, true labels of images were ignored. Instead, we threshold the pixels to be greater or smaller than $0.5$,

and labeled as $y_i = 0$ or $y_i = 1$. That is, we turned the grey-scaled values to a binary coding. Then, all pixels were described by a two-dimensional input in the range $[-1.0, 1.0]$, that indicates the coordinate of each output datum. In the case of the *zero* image, we splitted the data in four areas, i.e. the four corners, as is shown in the subfigure (A) of Figure 4. Each one of the modular tasks was initialized with an equally spaced grid of $M_k = 16$ inducing-inputs. The meta-GP required $M = 25$ in the case of the number *zero* and $M = 16$ for the *one*. The plotted curves correspond to the test GP predictive posterior at the probit levels $[0.2, 0.5, 0.8]$. The setup of the VEM algorithm was $\{\text{VE} = 20, \text{VM} = 10, \eta_m = 10^{-3}, \eta_L = 10^{-5}, \eta_\psi = 10^{-6}, \eta_Z = 10^{-5}\}$.

### F.2  Performance metrics

In our experiments, we used three metrics for evaluating the predictive performance of the global GP solutions: i) negative log-predictive density (NLPD), ii) root mean square error (RMSE) and iii) mean absolute error (MAE). Given a test input datum $\boldsymbol{x}_t$ and $\{\hat{f}_t, \hat{y}_t\}$ being the predictive mean of the GP function and output prediction respectively, the metrics can be computed as

$$\text{NLPD} = -\sum_{t=1}^{N_t} \log p(y_t|\mathcal{D}),$$

$$\text{RMSE} = \sqrt{\frac{1}{N_t}\sum_{t=1}^{N_t}(\hat{f}_t - f_t)^2},$$

$$\text{MAE} = \frac{1}{N_t}\sum_{t=1}^{N_t}\left|\hat{f}_t - f_t\right|,$$

where $y_t$ and $f_t$ are the true output target and function values. $N_t$ is the number of test data points.

### F.3  Optimization algorithms

Sometimes with sparse variational GP models, if one fits both hyperparameters $\boldsymbol{\psi}$, inducing points $\boldsymbol{Z}$ and variational parameters $\boldsymbol{\phi}$ at the same time, the optimization becomes difficult. This is mainly due to the hyperparameters $\boldsymbol{\psi}$ and $\boldsymbol{Z}$, whose abrupt changes while optimising may affect to the conditional GP densities and provide errors. To avoid this and also inspired in the well-known *expectation-maximization* (EM) algorithm and coordinate ascent methods, we alternatively can *freeze* and *unfreeze* hyperparameters and parameters during optimization. The following version of the variational expectation-maximization (VEM) algorithm was used mainly for the training of GP modules. That is the reason why we do not include the sub-scripts $\{k, *\}$ in the parameter variables.

---

**Algorithm 1** — VARIATIONAL EM FOR MODULAR GPS
___________________________________________________________________

1: Initialize $\boldsymbol{\psi}, \boldsymbol{\phi}$ and $\boldsymbol{Z}$
2: **while not** $\mathcal{L}_{\mathcal{M}}^{(t)} \approx \mathcal{L}_{\mathcal{M}}^{(t-1)}$ **do**
3:      # Variational Expectation (VE)
4:      **for** $j \in 1, \ldots, \text{VE}$ **do**
5:          update $\boldsymbol{\mu}_{(j)} \leftarrow \boldsymbol{\mu}_{(j-1)} + \eta_\mu \nabla_\mu \mathcal{L}_{\mathcal{M}}$
6:          update $\boldsymbol{L}_{(j)} \leftarrow \boldsymbol{L}_{(j-1)} + \eta_L \nabla_L \mathcal{L}_{\mathcal{M}}$
7:      **end for**
8:      # Variational Maximization (VM)
9:      # Hyperparameters
10:     **for** $j \in 1, \ldots, \text{VM}$ **do**
11:         update $\ell_{(j)} \leftarrow \ell_{(j-1)} + \eta_\psi \nabla_\ell \mathcal{L}_{\mathcal{M}}$
12:         update $\sigma_{a,(j)} \leftarrow \sigma_{a,(j-1)} + \eta_\psi \nabla_{\sigma_a} \mathcal{L}_{\mathcal{M}}$
13:     **end for**
14:     # Inducing-inputs
15:     **for** $j \in 1, \ldots, \text{VM}$ **do**
16:         update $\boldsymbol{Z}_{(j)} \leftarrow \boldsymbol{Z}_{(j-1)} + \eta_Z \nabla_Z \mathcal{L}_{\mathcal{M}}$
17:     **end for**
18: **end while**
___________________________________________________________________

For the distributed GP regression models needed for the baseline methods, we used the LBFGS optimization algorithm with a learning rate $\eta = 10^{-2}$. We set a default maximum of 50 iterations.

For the learning of meta-GPs we observed that the Adam optimizer (Kingma and Ba, 2015) was well-suited for this task. Details on learning rates and epochs can be found in the experiments included in our Github repository `https://github.com/pmorenoz/ModularGP`.

## G  Discussion on Assets

**F.1  License of scripts.**  Our code is based on the Pytorch framework, publicly available under the BSD license, and whose copyright is available at `https://github.com/pytorch/pytorch/blob/master/LICENSE`. We did not modify or redistributed the library, and its use was only for a research purpose. Additionally, we made use of the GPy library (`https://github.com/SheffieldML/GPy`) which is also under the BSD license. Finally, the code for the kernel using the Pytorch syntax is based on the original scripts of Steven Atkinson, which are publicly available at `https://github.com/cics-nd/gptorch` under the MIT license.

**F.2  Data consent and identifiable information.**  For our experiments, we used two datasets whose information comes from people. For both the London household and the US airlines datasets, the information was anonymous and pre-processed for guaranteeing the privacy. In the case of the London dataset, the *latitude-longitude* coordinates were normalized to the $[0, 1]$ range. In both cases, consent was given by authorities for using the data for statistical analysis in the public websites.

**F.3  Computing resources used.**  All experiments were carried out in *MacBook Pro (2021)* with an *M1* chip processor with 16GB of memory. No extra GPU resources were needed, neither for the 1.5M dataset.