# OpenReview forum: "Modular Gaussian Processes for Transfer Learning"
_NeurIPS.cc/2021/Conference — NeurIPS 2021 Poster_

### Official Review · Reviewer_2qsL · 2021-07-12

**Rating:** 7
**Confidence:** 5

**Summary:**

The authors propose a framework of reusing previously trained sparse variational GP models, without the need for revisiting the original data.
Each variational sparse GP model is called a "module".
The authors assume K modules, each of which is trained individually on a dataset $D_k$ (for $1 \leq k \leq K$) by means of ELBO maximisation (equation (1)).
The trained modules are subsequently combined into a meta-GP, which is defined as a global variational approximation of a large-dimensional GP.
This is achieved by deriving a global variational bound, where the global likelihood is approximated by utilising the local variational distribution in eq. (4).
Eventually, the evidence lower bound of eq. (6) is produced, which is optimised with respect to the global variational parameters $\phi = \{\mu, S\}$, as well as the hyperparameters $\psi $ and the (global) inducing point locations $Z$.
The framework is also extended to a multi-output context following MOGP.
The experimental campaign demonstrates that the methodology proposed is highly competitive on a number of large-scale datasets.

**Limitations And Societal Impact:**

In terms of societal impact, the authors comment on the need for protection of fitted models, as these may contain sensitive information.

**Main Review:**

The variational framework that is proposed to combine pre-trained GP modules into a single approximating model is well-motivated and novel, to the best of my knowledge.
The paper is well-written and the related work is adequately discussed.
The methodology proposed is demonstrated on a number of examples, while the numerical experiments confirm that the method is competitive to the state-of-the-art on a number of datasets, some of which involve millions of instances.
I think this work is of interest to the community, as it may have significant applications not only due to its scalability properties, but also due to the possibility of reusing trained models based on a rigorous framework.


The are however a few points that need to be clarified.

I found the discussion of the VEM algorithm (Algorithm 1) in the appendix somewhat confusing.
No EM algorithm has been discussed or motivated in the main paper.
Before reading the appendix, I had been under the impression that the variational parameters ($\mu$ and $L$), the hyperparameters $\psi$ and the inducing locations $Z$ and were being jointly optimised with respect to the ELBO in Equation (6).
In the joint optimisation case, $\psi$ and $Z$ are treated in terms of empirical Bayes (due to the ELBO).
However, according to the VEM algorithm, these three classes of parameters are optimised independently in three nested optimisation loops.
In particular, the optimisation of $\mu$ and $L$ is called as the E-step, followed by 2 M-steps for $\psi$ and $Z$, which are treated as latent variables.
I do not understand the motivation behind this choice.

In line 176, it is stated that the sum of expectations in (5) works as an average contrastive indicator.
I am not sure that I understand that "contrastive" means in this context.
Could you elaborate on this?

**Time Spent Reviewing:**

8

---

> ### Author Response · Authors · 2021-08-09
> **Response to Reviewer 2qsL**
>
> We thank the reviewer for the consideration, positive comments and recognition of the novelty of our paper. We are willing to clarify point-by-point the main questions included in the review.
>
> **Point 1**
> > I found the discussion of the VEM algorithm (Algorithm 1) in the appendix somewhat confusing. No EM algorithm has been discussed or motivated in the main paper. Before reading the appendix, I had been under the impression that the variational parameters ($\mu$ and $L$), the hyperparameters $\psi$ and $Z$ are treated in of empirical Bayes (due to the ELBO).
>
> The reviewer is absolutely right on this point, the discussion on the VEM should be clarified in the appendix and also include a new discussion in the main paper, if a revised version is allowed. The reason to include it in the appendix is that in the beginning of the development of this work, when first prototypes of the modular model were coded, we wanted to have some stability in the optimization process. Sometimes with sparse variational GP models, if one fits both hyperparameters $\psi$, inducing variables $Z$ and variational parameters $\{\mu, L\}$ at the same time, the optimization becomes difficult. This is mainly due to the hyperparameters and $Z$, whose abrupt changes while optimising may affect to the conditional GP and provide errors. To avoid this and also inspired in the well-known EM algorithm and coordinate ascent methods, we alternatively *freeze* and *unfreeze* hyperpameters and parameters during optimization, to avoid the optimizer to look for all the optimal values at the same time. This is what is expressed in the Algorithm 1 of the appendix.
>
> At a later time, when we had a larger experience with the code and experiments began to work particularly well, as is shown in the paper, we found that the Adam optimizer (Kingma and Ba, 2014) was well-suited for the framework without the need of using an EM-inspired algorithm in the optimization. That is, $\{\psi, Z, \mu, L\}$ could be also optimised at once. That's why we omitted references to the VEM in the main text.
>
> > However, according to the VEM algorithm, these three classes of parameters are optimised independently in three nested optimisation loops. In particular, the optimisation of $\mu$ and $\L$ is called as the E-step, followed by 2 M-steps for $\psi$ and $Z$, which are treated as latent variables. I do not understand the motivation behind this choice.
>
>  The definition of the algorithm as variational EM (VEM) is inspired in the methods of Beal (2003), particularly equations 2.17, and 2.18. There are also later versions of similar VEM algorithms, but considering variational Bayes, something that we did not do. We consider in this case the variational parameters $\mu, L$ to be the ones modelling the posterior distribution $q(u_*)$ of the random variables $z_*$, and hence included as E-step. On the other hand, we initially considered $Z, \psi$ as the hyperpamaters of the model, so that's why we called their optimization M-steps.
>
> **Point 2**
> > In line 176, it is stated that the sum of expectations in (5) works as an average contrastive indicator. I am not sure that I understand that "contrastive" means in this context. Could you elaborate on this?
>
>  We agree that this sentence should be clarified. When we wrote that the *"sum of expectations in (5) works as an average contrastive indicator"*, we wanted to say that Eq.\ (5) is the key expression that leads the variational distribution $q(u_*)$ to be similar to each modular GP $q(u_k)$. We say average because it is an expectation term and we say "contrastive" because the distribution $q(u_*)$ is recurrently compared with every $q(u_k)$ with the help of the integral $q_{\mathcal{C}}(u_k) = \int q(u_*)p(u_k|u_*)du_*$ in L172. We will look for a better synonym for contrastive. Thanks for pointing it out.
>
> **References**
>
> - D. P. Kingma and J. Ba, Adam: A Method for Stochastic Optimization, ICLR 2014.
>
> - M. J. Beal, Variational Algorithms for Approximate Bayesian Inference, PhD Thesis, University College London, 2003.

---

> > ### Comment · Reviewer_2qsL · 2021-08-26
> > **Is the VEM algorithm not obsolete?**
> >
> > I thank the authors for their rebuttal; most of my concerns have been addressed.
> >
> > I am not surprised that a joint optimisation of $\psi, Z, \mu, L$ (ie via Adam) works just as well, if not better.
> > However, doesn't that make the VEM algorithm obsolete?
> >
> > Which of the two settings are you using in the latest version?

---

> > > ### Author Response · Authors · 2021-08-26
> > > **Additional response to Reviewer 2qsL**
> > >
> > > We thank the reviewer for the positive consideration of our response.
> > >
> > > > However, doesn't that make the VEM algorithm obsolete?
> > >
> > > Right, since the Adam method is valid for the optimisation of $\psi, Z, \mu$ and $L$, somehow the VEM is obsolete due to it takes longer time for similar results.
> > >
> > > > Which of the two settings are you using in the latest version?
> > >
> > > In the latest version of our experiments, for the results included in the manuscript, we used the Adam method (Kingma et al., 2014) as the standard optimizer for both classification and regression experiments.
> > >
> > > We will clarify this very well in the appendix if a revised version of the paper is allowed. Additionally, we will remark in the appendix that the VEM could be used as an alternative if the tuning of the Adam optimizer becomes difficult for additional scenarios considered by the readers, i.e. different learning rates for inducing points, kernel hyperparameters, etc.
> > >
> > > **References**
> > >
> > > - D.P. Kingma and J. Ba, Adam: A Method for Stochastic Optimization, ICLR, 2014.

---

### Official Review · Reviewer_abDr · 2021-07-14

**Rating:** 9
**Confidence:** 4

**Summary:**

A novel approach to learn from multiple tasks (involving different partitions of predictor datasets and different types of target variables) that share the same type of input and domain. The method approximates and bounds the overall (cross-task) log-likelihood with ELBOs and posteriors of each task/module. Using inducing points posteriors, the data in each task can be summarized by inducing points of each task and is not required in later tasks. This is an appealing property in practice since it obviates the need to store all datasets of all tasks, and the training can be performed in a task-distributed fashion.

**Limitations And Societal Impact:**

The authors mentioned data retention its impact on privacy. The authors could discuss how inducing points encapsulate or hides the information about each individual data, and if the dataset in each task reflects properties of a user group, does the inducing point reveal specific information about this group?

**Main Review:**

To be able to learn first on a large dataset and then quickly transfer to other tasks of smaller datasets is critical for modern machine learning tasks. Thus, I think the problem addressed by the authors is of high importance. The method proposed is novel and interesting (with small caveats not fully exposed), combining inspirations from previous work to address the transfer learning problem. In particular, the transfer does not require storing the full datasets of previously encountered tasks, which is appealing to many practical problems. Further, this method can be extended to multi-output and multi-dimensional GP latent, and can be stacked on top of each other according to tasks splitting to obtain a hierarchical meta-GP model.

The experiments are thorough, covering small and large datasets. although I am very curious about the results in other well-controlled settings. Despite this small imperfection and other minor issues I list below, I really enjoyed reading this paper. The paper is also written fairly clearly with nice figures explaining the intuitions, but some descriptions are not intuitive to readers like me, so please provide more details.

I am not completely up-to-date with the literature in this area, but I have not seen any methods that are trivially similar to this proposed here. The benchmark methods compared in the experiments are also quite different in methodology (though distributed in different senses), some are significantly less well-developed than this work. This argues for the novelty of this work, and if it is true that this method is that novel, the authors should further highlight this new contribution.

Could the authors provide details to the questions below? If addressed well, I am willing to further raise my score.
1. How does the approximation in (4) impact the bound? This is probably the only loose end in this method, and I hope to see some analysis in this respect. Are there cases when this approximation is particularly bad so that the objective is no longer a bound on the likelihood? Are there any tricks used to ensure the quality of this approximation?
2. Although we do not need to retain all the data for each task, we do retain the inducing points to compute $q_k$ for (6) and (7). How does the relative size of the inducing set to the task data set affect the performance of this method?
3. In the toy concatenation experiment, I think it would be great to analyse all the moving parts of this method. What are the effects of
   1) inducing points and its approximate posterior,
   2) approximation made in (4)
   3) modular training of hyperparameters? Importantly, I'd love to see a comparison to the best solution possible without modular training, for example, a full posterior computed exactly without inducing points.
If it's not quite possible to clearly dissect the contribution of these factors, please explain why, as it is not immediately clear to me.
4. In the description of the banana dataset experiment, why is it not possible to control the number of inducing points? How would the two methods compare if this number is controlled?
4. All the figures need proper captions. I know the space is limited, but the figures make less sense and the readers have to guess.
5. The bold and normal Z's refer to unrelated objects, please consider using other symbols?
6. In line 229, what does $i$ index? Is this related to LMC?
7. In line 313, what is the exact inference solution? Exact in what sense?
8. In line 303, what is the "combination of solutions" referring to?
9. Figure 5 is not referred to.
10. There should at least be some discussions in the main text on training and optimization.

===== post rebuttle ======

I thank the authors for their reply. I also do not see serious technical concerns from other reviewers, except for the presentation and minor details (I now recall some initial difficulty in reading which is expressed in other reviews). This is a great paper and I raise my score and confidence. The value of this paper may be more obvious in the near future.

I would strongly encourage the authors to revise the paper to include the content of ALL their responses provided here.

**Time Spent Reviewing:**

3

---

> ### Author Response · Authors · 2021-08-09
> **Response to Reviewer abDr**
>
> We thank reviewer for the useful comments, positive consideration and relevant feedback on our paper. We appreciate that the main contributions and novelty of the work have been recognised. In the following lines, we will answer point-by-point to every question and comment, since we want to clarify and demonstrate the robustness of our method.
>
> **Question 1**
> > How does the approximation in (4) impact the bound? (...) of this approximation?
>
> We saw empirically that the likelihood approximation in (4) works particularly well for Gaussian and Bernoulli distributions in our experiments. We think that this is likely to be true for every exponential-family density, or at least if the likelihood model is assumed to be unimodal. In such case, if the variational density $q(u_k)$ is well-fitted to the main mode of the probability mass of the posterior distribution, the inverted rule will work well and the approximation will be close to the true likelihood distribution. If the posteriors $q(u_k)$ are not well-fitted, the inversion of the Bayes rule might not be effective, so the learning of the meta-GP could be erroneous. However, this has not been the case in our large amount of experiments so far. We will add further analysis on this approximation in the appendix if a revised version of the paper is allowed.
>
> **Question 2**
> > Although we do not need to retain all the data for each task, we do retain the inducing points to compute
> for (6) and (7). How does the relative size of the inducing set (...)
>
> Good point. Each $k$-th module $\mathcal{M}_k$ is composed by $\phi_k$, $\psi_k$, $Z_k$ and L_k.
>
> We observe that the higher the size of the subsets $Z_k$ is, the better the modular meta-GP, due to more variables are indicating to $q(u_*)$ how it should behave. In terms of memory storage, $Z_k$ is not a problem. The computation of the expectation terms is relatively cheap, but it is true that we have to invert matrices of size $M_k \times M_k$ (see Appendix, Section B), so the maximum order of $Z_k$ should be lower than $M_k = 1000$ to be affordable.
>
> **Question 3**
> > In the toy concatenation experiment, I think it would be great to analyse all the moving parts of this method. What are the effects of: (**A**) inducing points and its approximate posterior, (**B**) approximation made in (4) and (**C**) modular training of hyperparameters?
>
> (A) We did plenty of experiments with the toy dataset, and many insights were obtained from them. Particularly, we found that the meta-GP learning was robust to several locations and different configurations of the inducing variables. The input region is split up in five modules in Figure 4, but we also split it up in more than five sections for the experiments with 10M, 100M and 1M data points. Even with random initializations and different solutions of the variational posteriors $q(u_k)$, the final solution was well fitted. (B) Similarly as in the previous point, we saw that the approximation in (4) was robust and did not "suffer" neither with the number of inducing variables $M_k$ or the number of data-points per module $N_k$. (C) This is also a  good point to consider. It is true that several modular approximations might learn different hyperparameters $\psi_k$. Fortunately, as we have also mentioned in other points of the response, the meta-GP is fitted via an expectation term to all the modules. This makes the final hyperparameters $\psi_*$ to be on average close to all GP priors characterized by $(\psi_k)^K_{k=1}$, i.e.\ lengthscale and amplitude for the *vanilla* kernel. In the Appendix, section A.3, we show that the GP predictive $q_\mathcal{C}(u_k)$ is computed using the conditional GP prior given $\psi_*$.
>
> > Importantly, I'd love to see a comparison to the best solution possible without modular training, (...) it is not immediately clear to me.
>
> For the stochastic variational GP without the modular structure, we obtained an NLPD metric of $2.58\pm 0.11$ for $N=$10k in the toy dataset of Figure 1 and Table 2. We also trained a full GP with covariance inversion on 1000 data points from the same experiment, where we obtained an NLPD metric of $2.19 \pm 0.02$ under several initialisations. These NLPD values can be compared with the ones obtained in Table 2.
>
> **Question 4**
> > In the description of the banana dataset experiment, why is it not possible to control the number of inducing points? How would the two methods compare if this number is controlled?
>
> In the banana experiment, we saw that our performance on NLPD was better that the baseline variational sparse GP classification model. At the beginning, this improvement may seem odd, as we are using the modular framework. Later, we understood that even using the same number of inducing points for both the meta-GP and the baseline variational GP ($M_*=25$), the improvement was due to the modular GPs in our framework with $M_k = 9$ were also exploring better the data space.
>
> In summary, even if the meta-GP and the baseline have the same number of inducing points and hence, order of complexity,  the modules are also exploring with their own $Z_k$ inducing variables. So in the end, the modular framework is using more inducing variables to characterize the dataset than other standard variational sparse GPs.
>
> **Question 5**
> > All the figures need proper captions. I know the space is limited, but the figures make less sense and the readers have to guess.
>
> We agree that more information should be included in the captions for a better comprehension of the figures. We will do our best to make this possible.
>
> **Question 6**
> > The bold and normal Z's refer to unrelated objects, please consider using other symbols?
>
> That's true, we used both bold and normal Z to refer the normalisation constant and the subsets of inducing variables. We will use another letter for the normalization constant. Thanks for pointing it out.
>
> **Question 7**
> > In line 229, what does $i$ index? Is this related to LMC?
>
> The index $i$ in the equation $f_k(x) = \sum^Q_{q=1}\sum^{R_q}_{i=1}a^i_q v^i_q$ indicates the index of realization $i=1,2,\dots, R_q$ of each $v^{i}_q$ from the $q$-th latent process $u_q(\cdot)$. We took the notation of the LMC in Alvarez (2012). We will define this in the main paper if a revised version is allowed.
>
> **Question 8**
> > In line 313, what is the exact inference solution? Exact in what sense?
>
> The baseline GP methods BCM, PoE, GPoE and RBCM are based on exact inference for the GP regression model. That is, they compute the exact solution inverting the covariance matrices of the GP model for each one of the smaller tasks. When we said that we are closer to the *exact inference solution*, we refer that the performance of our framework based on variational approximations is similar to the four baseline methods.
>
> **Question 9**
> > In line 303, what is the "combination of solutions" referring to?
>
> Our method aims to fit a new variational distribution $q(u_*)$ without observing data, but fitting it to be very similar (in average) to every $k$-th modular solution $q(u_k)$. This learning method of the meta-GP based on an expectation term  is very robust to bad-fitted modules --- see Eq. (6). That is, if one of the $k$-th solutions $q(u_k)$ is not well-fitted, the final solution of $q(u_*)$ can be still very good due to the contribution of the rest of modules $k'\neq k$. However, baseline methods are not that robust to this scenario. Mainly, due to the meta-GP on those cases is an *explicit* combination of the posterior densities of the modular tasks. If we look to Deisenroth (2015) as an example, we will see that Eq. (20) in that paper has the following form
>
> $$p(f_*|x_*, \mathcal{D}) \propto \prod_{k=1}^{M}p_k(f_*|x_*, \mathcal{D}^{(k)}),$$
>
> where $p_k(f_*|x_*, \mathcal{D}^{(k)})$ is the posterior of the $k$-th task, equivalent to our $q(u_k)$. From the previous expression, one can see that if one of these $p_k(f_*|x_*, \mathcal{D}^{(k)})$ crashes, their meta-GP will also crash given that particular model. This fact was also proved in the implementation of the baselines during the experiments, that we will make public together with the modular GP package in Pytorch.
>
> **Question 10**
> > Figure 5 is not referred to.
>
> The reviewer is right, we will refer it in the (iv) experiment on L320--323, where there is still space.
>
> **Question 11**
> > There should at least be some discussions in the main text on training and optimization.
>
> We agree on this point, some discussion on the optimization should appear in the main text. We will do it if the revised version, if allowed. Initially, we included details on the training and optimization of hyperparameters and parameters in the main text, but the lack of space made us to move them to the appendix in the end. Sorry about that.
>
> **Limitations and Societal Impact**
> > The authors mentioned ...
>
> This is a good point of analysis in terms of privacy preservation for sparse GP approximations. We could imagine that on certain datasets, i.e.\ medical health records or digital registers from personal smartphones, the input data points could contain sensitive information about the patient or user, so we should try to protect them. It is true that the inducing points are themselves a form of hiding the true data, since they are model variables on the continuous space of the data. However, it is also true that they indicate the *main areas* of interest on such input data space, so privacy preserving methods should be considered for them (Smith, 2018; 2019). Another option would be to consider sparse GP approximations where the inducing variable do not lie directly on the input data space, for instance due to transformations. One example would be Van der Wilk (2017).
>
> **References**
> - M. T. Smith, et al. Differentially Private Regression with Gaussian Processes, AISTATS, 2018.
>
> - M. T. Smith et al. Differentially Private Regression and Classification with Sparse Gaussian Processes, arXiv 1909.09147, 2019.

---

### Official Review · Reviewer_FB5F · 2021-07-16

**Rating:** 6
**Confidence:** 4

**Summary:**

Summary: the paper proposes a modular decomposition of the conventional GP process. The author claims that such novel decomposition can empirically improve the prediction accuracy. However, the paper lacks certain theoretical reasoning behind the claims. Many critical definitions are missing and the presentation quality of the paper is below the NeuRIPS standard.

**Limitations And Societal Impact:**

Yes, the authors adequately addressed the limitations and potential negative societal impact of their work.

**Main Review:**

Weakness:

The presentation made the paper hard to follow, without a clear story line to seize. For example, why is it necessary to introduce the inducing variable $z_k$, and how are $z_k$ computed from D_k given x and y? Things get even worse when the subscript notation * is introduced, making the notations harder to interpret. Many notations are not defined，such as $L_k^*$, $f_{+\neq u_*}, V_{+ \neq *}$. Most importantly, the paper lacks discussions that why decompostion on the data is advantageous for GP learning. I listed some of my confusions below.

79-81 “consider that N_k for all k ∈ {1, 2, . . . , K} is sufficiently large for not accepting exact GP inference due to temporal and computational demands.” What does this sentence mean?

89-90 ”…being $u_k$ values of the same function $f(\cdot)$” (grammarly wrong).

144: notation $f_{+\neq u_*}$ is not defined here.

153: why $p(y|f_+)=\sum_k \log p(y_k|f_+)$? Are the marginal of $ p(y) $and $p(y_k)$ expected to be same (I think so, because k seems to be indexing a subsets of the entire data., i.e., samples of size $N_k$ of the entire data)? If so, then there should be an additional multiplicative 1/K constant on the r.h.s term.

178-179: “Without the need of revisiting any data, the GP predictive $q_C(u_k)$ is playing a different role in contrast with the usual one.” This sentence is seemingly trying to support the novelty claim, but failed to deliver the message. To be honest, the whole discussion paragraph that claims the advantage of such modular method (175-182) is difficult to interpret even if one with proper GP background is 100% focused. I spent hours on this paragraph but still cannot get the insights behind.

189: $L_k^*$ is not defined.

240: the reasoning behind such factorization is unclear, partly because the notation V_{+ \neq *} is undefined.

It leaves one wonder how the GP helps the transfer learning task in the empirical section at all. Please clarify this point. It seems to me the tasks are conventional Regression/Classification GP tasks, without the need of transfer learning. Please clarify how the advantage of the proposed method is reflected in the experiments section.

The usage of word "accept" has been abused everywhere.

**Time Spent Reviewing:**

6 hours

---

> ### Author Response · Authors · 2021-08-09
> **Response to Reviewer FB5F**
>
> We thank the reviewer for the useful comments and all the relevant feedback on the notation, we are willing to improve the main points to increase the clarity of our paper. We want also to remark, that the model is not exactly a decomposition of the conventional GP process to improve the prediction accuracy, as it is summarized in the review. Instead, it is a framework to build new GP models from others, without revisiting any data. We hope the following answers to each of the points raised by the reviewer can help to clarify our approach.
>
> **Point 1**
> > why is it necessary to introduce the inducing variable $z_k$, and how are $z_k$ computed from $D_k$ given x and y?
>
> The input-output dataset $\mathcal{D}$ of size $N$ is too big or it cannot be centralised on a single machine for learning. Therefore, it is divided into $K$ subsets $\mathcal{D}_k$ of size $N_k$. We would like to learn an exact GP on every $k$-th subset, but it would be very costly, due to the sizes $N_k\gg 1000$ and the inversion of the covariance matrix is $\mathcal{O}(N_k^3)$. Consequently, we used sparse approximations based on $M_k$ inducing inputs $z_k$, where $M_k\gg N_k$. Together with variational methods (Titsias, 2009; Hensman, 2015), we perform inference of the posterior GP on the function evaluations of such variables $z_k$.
>
> **Point 2**
> > Many notations are not defined, such as $L*_{k}$, $f_{+\neq u_*}$, $V_{+\neq*}$
>
> The lower bound $L*_{k}$ is presented in Eq. (1) and L106-107, we will add an extra definition when it is used in L189--192. We answer to the point on $f_{+\neq u_*}$ in **Point 5** (see below). The presentation of $V_{+\neq*}$ could be also improved in L240, we will do it and the notation is explained in **Point 9**.
>
> **Point 3**
> > 79-81 "consider that $N_k$ for all $k \in \{1,2,\dots,K\}$ is sufficiently large for not accepting exact GP inference due to temporal and computational demands." What does this sentence mean?
>
> The dataset is assumed to be divided into $K$ data *modules* or subsets of size $N_k$. This data size $N_k$ is still too large to train an exact GP, due to the inversion of the covariance matrix will take so much time when $N_k\gg 1000$. The computational cost would be $\mathcal{O}(N_k^3)$. This problem leads us to introduce variational inference methods and sparse GP approximations to train the models over each data *module* or subset of size $N_k$.
>
> **Point 4**
> > 89-90 "… being $u_k$ values of the same function $f(\cdot)$" (grammarly wrong)
>
> Sorry about the grammar mistake, we will rephrase it to be correct.
>
> **Point 5**
> > 144: notation $f_{+\neq u_{*}}$ is not defined here.
>
> The *augmented* GP is defined in L130, and the definition $p(f_+) = p(f_{+\neq u*}|u_*)p(u_*)$ in L144 refers to how some values of such GP can be conditioned on $u_*$ using the properties of Gaussian densities. The notation $f_{+\neq u_*}$ just indicated the values of such GP, not considering the ones used for conditioning, these are $u_*$. We will clarify this in the revised version, if allowed.
>
> **Point 6**
> > 153: why $p(y|f_{+}) = \sum_{k} \log p(y_k | f_+)$? Are the marginal $p(y)$ and $p(y_k)$ expected to be same (I think so, because k seems to be indexing a subsets of the entire data., i.e., samples of size $N_k$ of the entire data)?  If so, then there should be an additional multiplicative 1/K constant on the r.h.s term.
>
> There is a $\log(\cdot)$ operator missing in the reviewer's comment. In the submitted manuscript, the exact equation, written down in L153, is
>
> $$\log p(y|f_{+}) = \sum_{k=1}^{K} \log p(y_k|f_{+}),$$
>
> where $y = (y_k)^K_{k=1}$. Each $y_k$ is a subset of output data-points as is defined in L132. In L150, we explain that given the augmented stochastic process $f_{+}$, we are able to apply conditional independence (CI) across these subsets. Therefore the likelihood density factorises according to $p(y|f_{+}) = p(y_1, y_2, \dots, y_K|f_{+}) =  \prod_{k=1}^{K}p(y_k|f_{+})$ and if we consider the logarithm operator, it gives us the equation from L153.
>
> It is also important to mention that such factorization based on conditional independence is a well-known technique for the application of variational methods to sparse GP approximations. For instance, it is also used in Hensman (2013, 2015) to factorize the expectation terms and apply stochastic gradient descent accordingly.
>
> **Point 7**
> > 178-179: "Without the need of revisiting any data, the GP predictive $q_{C}(u_k)$ is playing a different role in contrast with the usual one.” This sentence is seemingly trying to support the novelty claim, but failed to deliver the message. To be honest, the whole discussion paragraph that claims the advantage of such modular method (175-182) is difficult to interpret even if one with proper GP background is 100\% focused. I spent hours on this paragraph but still cannot get the insights behind.
>
> We will clarify this paragraph to be more concise on the message. What we wanted to say is that the (variational) predictive posterior equation in GP models is typically computed in the literature as
>
> $$q(f_*) = \int p(f_*|u)q(u)du,$$
>
> where $f_*$ are the points of the function that we want to predict, $q(u)$ is the variational posterior already *learned* and $p(f_{*}|u)$ just the GP conditional prior density. However, in our case, the GP predictive $q_{\mathcal{C}}(u_k)$ comes from the equation included in L172, that is
>
> $$q_{\mathcal{C}}(u_k) = \int q(u_*)p(u_k|u_*)du_*,$$
>
> where $q(u_*)$ is the posterior distribution that *we want to learn* for the meta-GP and $p(u_k|u_*)$ is the GP conditional. We say that the previous predictive equation plays a different role due to $q(u_*)$ is not learned yet and we are using it to match the model with the modules. This is what we wanted to summarize in the paragraph L175--182.
>
> **Point 8**
> > 189: $\mathcal{L}^*_k$ is not defined.
>
> $\mathcal{L}^{*}_k$ refers to the $k$-th lower bound trained on each module. It is presented in Eq. (1) and L106-107, but we will add an extra definition in L189 when it is used to clarify.
>
> **Point 9**
> > 240: the reasoning behind such factorization is unclear, partly because the notation $V_{+ \neq *}$ is undefined.
>
> The reviewer is right that the notation of $V_{+ \neq *}$.
>
> The augmented set of latent functions $V$ could be decomposed as $V$ $=$ $(V_{+\neq*} \cup V_*)$, where $V_* = (v_{*q})^{Q}_{q=1}$. We did not want to make the reader get lost with the multi-output dense formulation. We will clarify this and explain the derivation carefully in the appendix, if a revised version is allowed.
>
> **Point 10**
> > It leaves one wonder how the GP helps the transfer learning task in the empirical section at all. Please clarify this point. It seems to me the tasks are conventional Regression/Classification GP tasks, without the need of transfer learning. Please clarify how the advantage of the proposed method is reflected in the experiments section.
>
> The use of the modular GP framework for transfer learning is based on the use of the multi-output GP model. In the experiments, we use the US-Flight dataset for this purpose (L328). Particularly, we imagine the case of having independent GP models trained on data from different days or months. If one wants to improve the performance on one particular day, it is not possible to incorporate more data from others, due to it cannot be accessed or was removed. Instead, what we can do is to build a meta-GP from the models trained on several days that were stored. To transfer the uncertainty from some models to others, the multi-output GP is specially well suited, as some *common* latent functions $\mathcal{V}$ are shared. In the end, we are building a multi-output GP from independent single-output GPs, without revisiting any data. The inner structure of the multi-output model allows to improve prediction on some outputs (days or months) using the knowledge learned from others.
>
> **Point 11**
> > The usage of word "accept" has been abused everywhere.
>
> We see that it has been used six times in the nine pages of the manuscript. We will use synonyms instead in the revised version of the paper, if allowed.
>
> **References**
>
> - M. K. Titsias, Variational Learning of Inducing Variables in Sparse Gaussian
> Processes, AISTATS 2009.
>
> - J. Hensman, N. Fusi and N. D. Lawrence, Gaussian Processes for Big Data, UAI 2013.
>
> - J. Hensman, A. G. de G. Matthews and Z. Ghahramani, Scalable Variational Gaussian Process Classification, AISTATS 2015.

---

> > ### Comment · Reviewer_FB5F · 2021-08-16
> > **Follow up comments**
> >
> > Many thanks to the authors for explaining the missing definitions on certain notations. Those explanations indeed helped to clarify many important connections between the derived equations. I would also appreciate it if the updated version can better organize and explain the notations especially when there is subscript present, since this paper involves many variables. I believe the authors have clarified some misunderstandings and addressed my main concerns in their response, especially on the missing definitions. I correspondingly increase my score to 6.

---

### Official Review · Reviewer_ntYs · 2021-07-16

**Rating:** 6
**Confidence:** 3

**Summary:**

The paper proposes an ensemble GP method (where each model is termed a 'module') that can fuse a set of pre-existing GPs to train a new "meta-GP".  The authors also present a closed form ensemble lower bound that considers the parametrisations of the 'modules'.

**Main Review:**

I rather enjoyed the motivating dialogue and examples.

"Each module is considered to be a sparse variational GP, containing only variational parameters and hyperparameters." Is it a necessary condition that each module be specifically a sparse variational GP? Could this approach be extended/directly applied to spectral approximations like in [R1]? I ask because given the generality of the proposal "We present a framework based on modular Gaussian processes" it is not so clear that the method applies only to a particular _class_ of GP derivatives (of which sparse vi methods are just one), or a more general class of GP approximations (of which, e.g., [R1] is also one).

Clarity:
The introduction starts with a rather involved sentence: "We develop a module-based method...revisiting any data". The sentence is rather convoluted and I would suggest condensing or splitting it up.

"n. However, GP models are not immune to settings where we need to adapt to irregular ways of observing data, e.g. asynchronous samples or missing inputs." I'm not so sure this is true (or at least the way it is phrased here). GPs are typically derived on continuous domains; i.e. ones that do not need data to be 'gridded' (although approximate methods have been proposed to accommodate structured data such as KISS-GP).

Related work:
I think [R0] should be included in the related work since it is addressing a similar problem but from the optimal transport perspective and applies in a potentially more general sense than KL divergence between stochastic processes.
Also, the approach of a "meta" GP seems directly related to the classic concept of model stacking. This has been attempted in the past (e.g. [R2]) and has also been termed a meta-gp. Could the authors comment on the relation to such prior work?

Compression: It seems to me that using a sparse VI gp, there is a consequent loss of information. If it was a full GP, there would be no information loss in the sense of the full covariance (of the 'past') data.

[R0]  Mallasto, Anton, and Aasa Feragen. "Learning from uncertain curves: The 2-Wasserstein metric for Gaussian processes." Advances in Neural Information Processing Systems. 2017.
[R1] Lázaro-Gredilla, Miguel, et al. "Sparse spectrum Gaussian process regression." The Journal of Machine Learning Research 11 (2010)
[R2] Neumann, Marion, et al. "Stacked Gaussian process learning." 2009 Ninth IEEE International Conference on Data Mining. IEEE, 2009.

**Time Spent Reviewing:**

4

---

> ### Author Response · Authors · 2021-08-09
> **Response to Reviewer ntYs**
>
> We thank the reviewer for the positive consideration and the useful comments on our work, particularly those ones related to references in the literature. In the following lines we will answer point-by-point to all reviewer's comments.
>
> **Question 1**
> > Is it a necessary condition that each module be specifically a sparse variational GP? Could this approach be extended/directly applied to spectral approximations like in [R1]?  I ask because given the generality of the proposal, it is not clear that the method applies only to a particular class of GP derivatives (...), or a more general *class* of GP approximations (...).
>
> Good question. In principle, the modular framework was developed for variational methods for sparse GP approximations based on inducing-points. When we initially claimed the generality of the proposal, we thought more on all the variants of Titsias (2009) from the last 10 years, i.e. Hensman (2013, 2015) or Saul (2016), to name a few. However, when we obtained the final expression of the module-driven lower bound in Eq. (6), we saw that it could also be general for multi-task GPs (Alvarez, 2008) and heterogeneous likelihoods (Moreno, 2018). Currently, we think that the framework also accepts sparse approximations based on more complex transformations of the inducing-variables as, for instance, the work in Van der Wilk, (2017) that introduces convolutional methods, which falls a bit out of the initial scope of the paper. With respect to the work of Lazaro-Gredilla (2010), our intuition says that it would be likely to be introduced as a module, but the variational inference scheme should be derived previously, something that was not done in their original JMLR paper.
>
> **Clarity Point**
> >  The introduction starts with a rather involved sentence: "We develop a module-based method...revisiting any data". The sentence is rather convoluted and I would suggest condensing or splitting it up.
>
> The reviewer is absolutely right on this 3-lines sentence in the abstract, we will split it up to be clearer and more concise if a revised version of the paper is allowed.
>
> **Irregular data**
> > However, GP models are not immune to settings where we need to adapt to irregular ways of observing data, e.g. asynchronous samples or missing inputs." I'm not so sure this is true (or at least the way it is phrased here). GPs are typically derived on continuous domains; i.e. ones that do not need data to be 'gridded' (although approximate methods have been proposed to accommodate structured data such as KISS-GP)
>
> We appreciate that the reviewer pointed this sentence out, it should be clarified. When we refer to irregular ways of observing data, it is not related to the continuous domain of input observations, but more to the availability of data samples, i.e. centralisation in one machine, or having subsets stored in different devices, with several data sizes. The accommodation of the data structure, particularly in the input domain is well addressed in KISS-GP for avoiding grids of inducing points, the reviewer is absolutely right. However, we have not focused on that particular problem in our paper.
>
> **Related work**
> > I think [R0] should be included in the related work since it is addressing a similar problem but from the optimal transport perspective and applies in a potentially more general sense than KL divergence between stochastic processes.
>
> We agree that Mallasto (2017) should be included in the related work, as they were able to incorporate uncertainty in a larger population analysis using 2-Wasserstein metrics. We will incorporate it in the revised version of the paper, if allowed. Additionally, it would be interesting to study the connection of our variational modular framework with optimal transport in further work.
>
> > Also, the approach of a "meta" GP seems directly related to the classic concept of model stacking. This has been attempted in the past (e.g. [R2]) and has also been termed a meta-gp. Could the authors comment on the relation to such prior work?
>
> The reviewer is right, there exists a connection of our work with the previous ideas of model stacking. As far as we understand the paper, the method of Neumann (2009) builds a meta GP enhaced by posterior covariance functions from other tasks. This idea is an earlier version of Deisenroth (2015) and also similar to the Bayesian Committee Model (BCM) of Tresp (2000) with exact GP regression tasks, that we included as part of the baseline methods in our experiments. We will also include a reference to this paper in the revised version of the manuscript, if allowed.
>
> **Compression Point**
> > It seems to me that using a sparse VI gp, there is a consequent loss of information. If it was a full GP, there would be no information loss in the sense of the full covariance (of the 'past') data.
>
> This is a very interesting point. Of course, the introduction of sparse VI GPs induces a slight loss of information, or at least some error due to the approximation. This is hardly avoidable if we want the GP model to scale up to thousands of data points and non-Gaussian likelihood models at the same time. Particularly, we wanted to show this on the experiments and make this clear to the reader, so we included in Table 3 the relative percentage of loss of information w.r.t. the fitted models before training the meta-GP. In the case of introducing only exact GPs without any sparse approximation, the practitioner could consider the work of Tresp (2000) and Deisenroth (2015), that we used to compare the performance of our method.
>
> One last detail that could be of interest is that our meta-GP bound is based on a expectation operator, so the meta-GP tries to be fitted *on average* to all the modular tasks. If one of them is bad-fitted, the final meta-GP is robust to this error. This is not the case for Deisenroth (2015) and exact GP models, because the final *meta* posterior GP is an explicit combination of the posterior densities of the tasks. This can be seen from Eq. (20) in that paper, that is,
>
> $$p(f_*|x_*, \mathcal{D}) \propto \prod_{k=1}^{M}p_k(f_*|x_*, \mathcal{D}^{(k)}),$$
>
> where $p_k(f_*|x_*, \mathcal{D}^{(k)})$ are the tasks, similarly to our modular models. Consequently, if one $k$-th GP crashes in this case, the meta-GP will also crash given the model of Deisenroth (2015).
>
> **References**
>
> - M. K. Titsias, Variational Learning of Inducing Variables in Sparse Gaussian
> Processes, AISTATS 2009.
>
> - J. Hensman, N. Fusi and N. D. Lawrence, Gaussian Processes for Big Data, UAI 2013.
>
> - J. Hensman, A. G. de G. Matthews and Z. Ghahramani, Scalable Variational Gaussian Process Classification, AISTATS 2015.
>
> - A. D. Saul, J. Hensman, A. Vehtari and N. D. Lawrence, Chained Gaussian Processes, AISTATS 2016.
>
> - P. Moreno-Muñoz, A. Artés and M. Alvarez, Heterogeneous Multi-output Gaussian Process
> Prediction, NeurIPS 2018.
>
> - M. Alvarez and N. Lawrence, Sparse Convolved Gaussian Processes for Multi-output Regression, NIPS 2008
>
> - M. Van der Wilk, C. E. Rasmussen and J. Hensman, Convolutional Gaussian Processes, NIPS 2017.
>
> - M. Lazaro-Gredilla, J. Quiñonero-Candela, C. E. Rasmussen and A. R. Figueiras-Vidal. Sparse Spectrum Gaussian Process Regression, JMLR 2010.
>
> - A. G. Wilson and H. Nickisch, Kernel Interpolation for Scalable Structured Gaussian Processes (KISS-GP), ICML 2015.
>
> - A. Mallasto and A. Feragen, Learning from uncertain curves: The 2-Wasserstein metric for Gaussian processes. NIPS 2017.
>
> - M. Neumann, K. Kersting, Z. Xu and D. Schulz. Stacked Gaussian process learning. IEEE International Conference on Data Mining. IEEE, 2009.
>
> - M. P. Deisenroth and J. W. Ng, Distributed Gaussian Processes, ICML, 2015.
>
> - V. Tresp, A Bayesian Committee Machine, Neural Computation, 2000.

---

### Author Response · Authors · 2021-08-09
**Response to Reviews**

We thank all reviewers for their useful comments, positive consideration and relevant feedback on our paper. We believe that there has been a general comprehension of our work, and particularly of the main contributions and novelty. To provide more clarity to all reviewers, we have answered to every question and comment included in the reviews. Moreover, we are interested to show the high robustness of the modular framework, particularly on the learning process of the meta-GP under multiple scenarios included in the experiments.

---

### Decision · Program_Chairs · 2021-09-27

**Decision:**

Accept (Poster)

**Comment:**

This paper proposes a framework for reusing previously learned sparse variational GP models without revisiting any data and a further extension to the multi-output context. All four knowledgeable reviewers recommend acceptance and I agree with them. As a clarification to the authors, yes, you are allowed and encouraged to incorporate amendments/additions to address the reviewers feedback in the final version of the paper.